# Any-Order GPT as Masked Diffusion Model: Decoupling Formulation and Architecture

## Abstract

Large language models (LLMs) predominantly use autoregressive (AR) approaches, but masked diffusion models (MDMs) are emerging as viable alternatives. A key challenge in comparing AR and MDM paradigms is their typical architectural difference: AR models are often decoder-only, while MDMs have largely been encoder-only. This practice of changing both the modeling paradigm and architecture simultaneously makes direct comparisons unfair, as it's hard to distinguish whether observed differences stem from the paradigm itself or the architectural shift. This research evaluates MDMs within a decoder-only framework to: (1) equitably compare MDM (as Any-Order AR, or AO-AR) and standard AR paradigms. Our investigation suggests that the standard AO-AR objective, which averages over all token permutations, may benefit from refinement, as many permutations appear less informative compared to the language's inherent left-to-right structure. (2) Investigate architectural influences (decoder-only vs. encoder-only) within MDMs. We demonstrate that while encoder-only MDMs model a simpler conditional probability space, decoder-only MDMs can achieve dramatic generation speedups ($\sim 25\times$) and comparable perplexity with temperature annealing despite modeling a vastly larger space, highlighting key trade-offs. This work thus decouples core paradigm differences from architectural influences, offering insights for future model design.

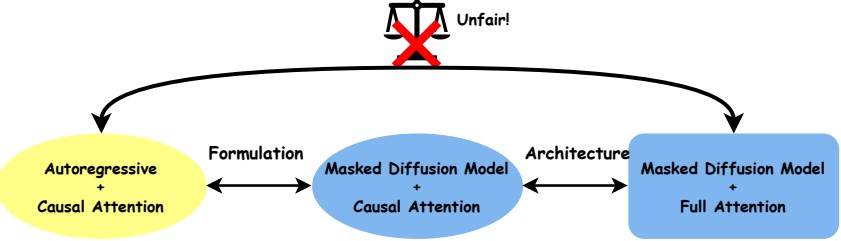

## 1 Introduction

The advent of large language models (LLMs) has fundamentally reshaped the field of language modeling, achieving remarkable success and quickly solidifying their position as the prevailing approach for tackling complex language tasks. While the prevailing methodology for state-of-the-art language modeling remains autoregressive (AR), focusing on next-token prediction, the remarkable success of diffusion models (Sohl-Dickstein et al., 2015; Ho et al., 2020; Song et al., 2020) in continuous domains (e.g., image generation) has naturally spurred investigation into the viability and potential benefits of discrete diffusion models (Austin et al., 2021; Campbell et al., 2022; Lou et al., 2023) for language modeling. Among the strategies emerging within the discrete diffusion framework, masked (absorbing state) diffusion models (MDMs) have captured considerable attention. Underscoring their potential, recent works such as LLaDA (Nie et al., 2025) and Dream (Ye et al., 2025) have successfully scaled these masked diffusion models to an impressive 7B parameters, achieving compelling results that challenge the sole dominance of AR methods.

Yet, in comparisons between AR and MDM paradigms, a fundamental *architectural difference* is often overlooked: the former typically operates within a *causal attention, decoder-only* setting, whereas the latter, as seen in recent works, commonly employs *full attention* within an *encoder-only* setup. Beyond the high-level paradigm, the choice between causal and full attention dictates significant differences in training, density estimation, and generation efficiencies. Intriguingly, recent studies such as ENTP (Ewer et al., 2024) and MAR (Li et al., 2024) provide evidence hinting that

full attention, despite its typical association with non-causal tasks, might hold inherent theoretical advantages and achieve superior empirical results even in contexts aiming for sequential, AR-like generation. This highlights the need to decouple the effects of the theoretical formulations (AR vs. MDM) from the underlying attention mechanism.

Building upon this observation, we propose and evaluate MDMs (as Any-Order AR) implemented within a decoder-only framework to enable a more equitable comparison against conventional autoregressive models. This setup allows us to isolate variables and answer the following two questions:

- Given the same decoder-only architecture, what are the fundamental differences in modeling capabilities and empirical performance between the standard AR formulation and the AO-AR formulation for language tasks?
- For Masked Diffusion Models on language tasks, what are the theoretical and empirical differences using an encoder-only versus a decoder-only architecture?

To answer these questions, we specifically design AO-GPT[1], a decoder-only architecture capable of modeling any-order sequences, and conduct a series of carefully designed experiments. In Section 3, we delve into the first question by systematically investigating the impact of varying the distribution over token prediction orders during training, while keeping the decoder-only architecture constant. Our analysis in this section, leading to Findings 1-3, aims to clarify some intrinsic characteristics of language, convergence properties, and the influence of different orderings. Subsequently, Section 4 addresses the second question by conducting a comparative analysis of encoder-only versus decoder-only architectures when implementing the MDM formulation. This investigation examines fundamental theoretical differences in how these architectures model conditional probabilities, their empirical performance on perplexity benchmarks, and crucial practical aspects such as generation efficiency, including computational complexity and inference speed. The insights from this section, encapsulated in Findings 5-8, are intended to clearly delineate the respective strengths, weaknesses, and trade-offs associated with each architectural choice for MDMs.

By looking closely at these aspects, this work aims to help better understand how different modeling approaches (AR vs. MDM) and model structures (decoder-only vs. encoder-only) work together. We think these findings will be helpful for both communities, for guiding future research and development in discrete diffusion and language modeling.

## 2 PRELIMINARY

### 2.1 BACKGROUND

**Autoregressive (AR) Models** factorize the data's likelihood using the chain rule from left to right:

$$-\log p_{\boldsymbol{\theta}}\left(\boldsymbol{x}\right) = -\sum_{i=1}^{n} \log p_{\boldsymbol{\theta}}\left(\boldsymbol{x}_i | \boldsymbol{x}_{<i}\right) \coloneqq \mathcal{L}_{\text{AR}}. \tag{1}$$

**Any-Order Autoregressive (AO-AR) Models,** unlike standard AR models which rely on a fixed factorization order, aim to model the likelihood of averaging or marginalizing over all possible $n!$ permutations of the data sequence. Prominent examples include NADE (Uria et al., 2016), XL-Net (Yang et al., 2019), Autoregressive Diffusion Models (Hoogeboom et al., 2021), and $\sigma$-GPT (Pannatier et al., 2024). The log-likelihood can be expressed as:

$$-\log p_{\boldsymbol{\theta}}\left(\boldsymbol{x}\right) = -\log \mathbb{E}_{\sigma \sim \mathcal{U}(S_n)} p_{\boldsymbol{\theta}}\left(\boldsymbol{x} | \sigma\right)$$

$$\leq \mathbb{E}_{\sigma \sim \mathcal{U}(S_n)}\left[-\sum_{i=1}^{n} \log p_{\boldsymbol{\theta}}\left(\boldsymbol{x}_{\sigma_i} | \boldsymbol{x}_{\sigma_{<i}}\right)\right] \coloneqq \mathcal{L}_{\text{AO-AR}}. \tag{2}$$

**Discrete Diffusion Models.** Among the various approaches within the discrete diffusion framework, MDMs (also known as absorbing state diffusion models) have gained significant attention. These models define a forward noising process where tokens are progressively masked:

$$q_{t|0}\left(\boldsymbol{x}_t | \boldsymbol{x}_0\right) = \prod_{i=1}^{n} q_{t|0}\left(\boldsymbol{x}_t^i | \boldsymbol{x}_0^i\right) = \prod_{i=1}^{n} \text{Cat}\left(\boldsymbol{x}_t^i; (1-t)\delta_{\boldsymbol{x}_0^i} + t\delta_{[\text{MASK}]}\right). \tag{3}$$

---

[1]In this paper, AO-AR refers to the generative formulation, which can be implemented with either encoder-only or decoder-only architectures. In contrast, AO-GPT specifically denotes our model that combines the AO-AR objective with a decoder-only architecture.

Here, $t \in [0, 1]$ represents the diffusion time (or masking level), interpolating between the original data $\boldsymbol{x}_0$ ($t = 0$) and a fully masked sequence ($t = 1$). D3PM (Austin et al., 2021) and CTMC (Campbell et al., 2022) follow the approach of DDPM (Sohl-Dickstein et al., 2015; Ho et al., 2020) to learn the posterior distribution $p_{0|t}(\boldsymbol{x}_0|\boldsymbol{x}_t)$ through maximizing the Evidence Lower Bound. SEDD (Lou et al., 2023) formulates discrete diffusion models using the likelihood ratio $\frac{p_t(\boldsymbol{x})}{p_t(\boldsymbol{y})}$, and using denoising score entropy to learn the likelihood ratio.

More recently, studies by MDLM (Shi et al., 2024; Sahoo et al., 2024) and RADD (Ou et al., 2024) have shown that for masked diffusion models, different parameterizations are equivalent, and the training objective can be simplified or directly derived from the likelihood. This leads to the following objective function, which is an ELBO on the data likelihood:

$$-\log p_{\boldsymbol{\theta}}(\boldsymbol{x}) \leq \int_0^1 \frac{1}{t} \mathbb{E}_{q_{t|0}(\boldsymbol{x}_t|\boldsymbol{x}_0)} \left[ \sum_{i: \boldsymbol{x}_0^i = [\text{MASK}]} -\log p_{\boldsymbol{\theta}}(\boldsymbol{x}_0^i|\boldsymbol{x}_t) \right] \mathrm{d}t \coloneqq \mathcal{L}_{\text{MDM}}. \qquad (4)$$

$\mathcal{L}_{\text{MDM}}$ and $\mathcal{L}_{\text{AO-AR}}$ have been shown to be equivalent through simple derivations using techniques in NADE (Uria et al., 2016) and RADD (Ou et al., 2024). In the remainder of the paper, we will use the terms "Masked Diffusion Models" and "Any-Order Autoregressive Models" interchangeably.

## 2.2 DECOUPLING FORMULATION AND ARCHITECTURE

**Training Efficiency.** Decoder-only models (Causal LM) leverage almost every token for prediction, offering high signal density. Encoder-only MDMs, while predicting more tokens (on average, 50%) than traditional Masked LM ($15\% \sim 25\%$), still typically utilize fewer tokens than AR.

**Density Estimation Efficiency.** To evaluate a joint density of a sequence under a given order (e.g, left-to-right), A decoder-only model computes sequence likelihood in a single pass ($O(n)$ complexity)[2]. In contrast, an encoder-only model requires $n$ separate network evaluations, leading to $O(n^2)$ complexity.

**Generation Efficiency.** Inference procedures also differ markedly. Standard decoder-only AR models generate $n$ tokens through $n$ sequential forward passes; efficient Key-Value (KV) caching makes the total complexity approximately $O(n)$. In contrast, encoder-only MDMs require $T$ iterative refinement steps. With each step processing the full sequence via full attention ($O(n)$ per step), their total complexity is around $O(T \cdot n)$. Notably, $T$ (the number of steps) is often comparable to $n$, partly due to conditional independence assumptions when generating multiple tokens simultaneously (Liu et al., 2024; Xu et al., 2024). For comparison, decoder-only masked diffusion models can also achieve $O(n)$ complexity.

These fundamental differences in training, density estimation, and inference characteristics across architectures highlight the critical need to decouple the generative formulation (AR vs. MDM/AO-AR) from architectural choices (causal decoder vs. full-attention encoder). Without such decoupling, comparing a standard decoder-only AR model to an encoder-only MDM inevitably conflates these two variables, obscuring a fair assessment of each paradigm.

## 2.3 EXPERIMENT SETTING

Our approach trains MDMs using a decoder-only AR framework. A fundamental requirement for this is enabling the model to predict tokens in an arbitrary, non-sequential order, a departure from standard left-to-right autoregression. To achieve this order-agnostic capability, we build upon the $\sigma$-GPT architecture (Pannatier et al., 2024), which integrates explicit target position information to guide predictions. We selected $\sigma$-GPT as our foundation over alternatives like XL-Net (Yang et al., 2019) due to its architectural alignment with contemporary decoder-only large language models.

Building on this baseline, our AO-GPT incorporates several key enhancements. We explore and refine methods for injecting target position information more effectively within the Transformer blocks and investigate the impact of training techniques such as Exponential Moving Average (EMA) to improve stability and performance. The inference mechanism for the diffusion steps also leverages an efficient sampling algorithm (Lemma 1) to minimize computational costs.

---

[2]For simplicity, our complexity analysis assumes the context length $n_{\text{ctx}}$ is relatively small compared to $12 \cdot d_{\text{model}}$, leading to a linear scaling with the number of tokens, as discussed in (Kaplan et al., 2020).

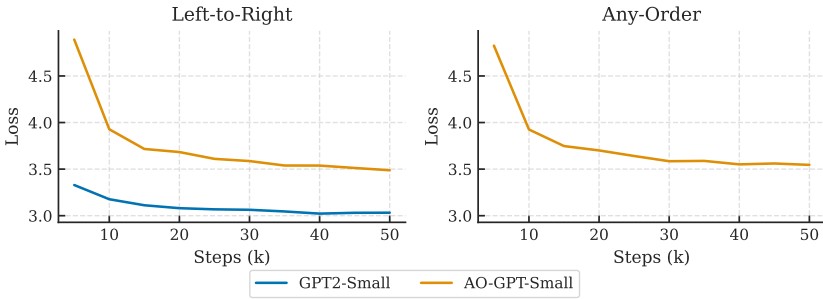

Figure 1: Training loss curves comparing a standard AR GPT against an AO-GPT. Both models employ an identical decoder-only architecture, with AO-GPT demonstrating slower initial convergence.

A comprehensive exposition of the AO-GPT architecture, its training paradigm, the specific design choices and ablations (including target position injection strategies and EMA), and the inference procedure is detailed in Appendix A.

Following SEDD, we train our models on the OpenWebText dataset (Gokaslan & Cohen, 2019), as the original WebText dataset is not publicly available. For evaluation, we test on a suite of standard benchmarks: LAMBADA (Paperno et al., 2016), WikiText2 (Merity et al., 2016), PTB (Marcus et al., 1993), WikiText103 (Merity et al., 2016), and the One Billion Words dataset (Chelba et al., 2013). A context length of 1024 tokens is utilized for all experiments. For data splits and processing, we exactly follow the methodologies outlined in SEDD, which includes techniques such as packing sentences to generate uniform-length input blocks.

## 3 AR VS. ANY-ORDER AR (MDM): A DECODER-ONLY COMPARISON

As established in the preliminaries and further elaborated by NADE (Uria et al., 2016) and recent analyses such as RADD (Ou et al., 2024), the training objective for MDMs ($\mathcal{L}_{\text{MDM}}$) is equivalent to that of Any-Order AR models ($\mathcal{L}_{\text{AO-AR}}$). This equivalence can be formally expressed as:

$$
\begin{aligned}
\mathcal{L}_{\text{MDM}} &= \int_0^1 \frac{1}{t} \mathbb{E}_{q_{t|0}(\boldsymbol{x}_t|\boldsymbol{x}_0)} \left[ \sum_{i:\boldsymbol{x}_0^i=[\text{MASK}]} -\log p_{\boldsymbol{\theta}}(\boldsymbol{x}_0^i|\boldsymbol{x}_t) \right] \mathrm{d}t \\
&\overset{\text{(Ou et al., 2024)}}{=} n \cdot \mathbb{E}_{l \sim U(1,\dots,n)} \frac{1}{n-l+1} \mathbb{E}_{\sigma \sim \mathcal{U}(S_n)} \sum_{r=l}^{n} -\log p_{\theta}(\boldsymbol{x}_{\sigma_r}|\boldsymbol{x}_{\sigma_{<l}}) \\
&\overset{\text{(Uria et al., 2016)}}{=} \mathbb{E}_{\sigma \sim \mathcal{U}(S_n)} \left[ \sum_{i=1}^{n} -\log p_{\boldsymbol{\theta}}\left(\boldsymbol{x}_{\sigma_i}|\boldsymbol{x}_{\sigma_{<i}}\right) \right] = \mathcal{L}_{\text{AO-AR}}
\end{aligned}
\tag{5}
$$

This mathematical equivalence is pivotal. It implies that when MDMs (via their AO-AR formulation) and standard AR models are implemented using the same decoder-only architecture, the fundamental difference in their training objectives lies in the distribution over token orders. Standard AR models adhere to a fixed, left-to-right order (i.e., the permutation $\sigma$ is the identity, $\sigma = \text{id}$), while AO-AR models (and thus MDMs) effectively learn by averaging over all possible $n!$ permutations ($\sigma \sim \mathcal{U}(S_n)$). Therefore, the central goal of this section is to investigate the following key question:

> **Question 1:** *Given the same decoder-only architecture, what are the fundamental differences in modeling capabilities and empirical performance between the standard AR formulation and the AO-AR formulation for language tasks?*

To address Question 1, we first examine the training dynamics of these two approaches when implemented within an identical decoder-only architecture, specifically focusing on their convergence behavior. We train a standard left-to-right AR model and an AO-AR model on the same OpenWebText (Gokaslan & Cohen, 2019) dataset, with exactly the same architecture and model size (standard GPT-2 Small size), and observe their loss progression. Specifically, we simultaneously compare the AR loss eq. (1) and AO-AR loss eq. (2). In the figure, these are labeled 'Left-to-Right' and 'Any-Order', respectively. Our initial experiments, illustrated in Figure 1, reveal a notable difference in training progression:

> **Finding 1:** *Any-Order GPT converges significantly slower in the initial training stages compared to its standard GPT counterpart when both utilize the same architecture.*

This initial observation (Finding 1) suggests that the any-order objective ($\sigma \sim \mathcal{U}(S_n)$), despite its flexibility, might slow initial learning due to increased task complexity compared to the fixed left-to-right ($\sigma = \text{id}$) order. The slower AO-AR convergence could arise from two main factors: (1). *Weight Sharing Burden*: learning representations effective across $n!$ permutations is demanding. (2). *Prevalence of Less Informative Orders*: many permutations in $\mathcal{U}(S_n)$ might be uninformative or act as noise.

To further probe the effect of prediction order, we next evaluate model training under conditions where a single, predetermined order is maintained throughout. We compare models trained on: a) conventional left-to-right ($\sigma = \text{id}$), b) a singular, randomly sampled permutation ($\sigma_{\text{rand}} \in S_n$) that remains fixed for the entirety of the training phase, and c) a block-wise permutation strategy, serving as an intermediate approach. Globally, this method processes blocks of tokens sequentially from left to right. However, within each block, tokens are processed according to a fixed, non-left-to-right permutation. For example, if the sequence is divided into 4-token blocks, the first block of tokens (indices 0,1,2,3) would be processed in the order $0 \to 2 \to 3 \to 1$. The next block (indices 4,5,6,7) would then be processed as $4 \to 6 \to 7 \to 5$. This specific intra-block permutation remains constant throughout training. Figure 2(a) presents these results. The comparison in Figure 2(a) leads to our second finding:

> **Finding 2.1:** *Even when both are trained on only one fixed order, the standard Left-to-Right order converges much faster than an arbitrary, randomly selected fixed order from $S_n$.*
> **Finding 2.2:** *Fixed block-wise random serves as an interpolation between Left-to-Right and purely random order in terms of convergence speed.*

**Remark 1** (Identical Loss Lower Bound). *The minimum achievable cross-entropy loss for any autoregressive factorization is the true entropy of the data, $H(\boldsymbol{x})$. By the chain rule, $H(\boldsymbol{x}) = \sum_{i=1}^{n} H(\boldsymbol{x}_{\sigma_i} | \boldsymbol{x}_{\sigma_{<i}})$ for any permutation $\sigma \in S_n$. Thus, the optimal loss value is identical regardless of the chosen generation order (e.g., left-to-right, any fixed random order, or averaged over all orders as in AO-AR). Differences in convergence or final empirical loss values therefore reflect practical learning challenges and inductive biases under different ordering schemes, not a difference in the achievable target for a perfect model. This ensures the fairness of our loss comparisons.*

This underscores language's intrinsic left-to-right structure. While order-agnosticism (and thus MDMs) offers flexibility, averaging over all permutations may be less optimal. Given the observed slower convergence of purely Any-Order models (Finding 1) and the apparent advantage of the L2R order (Finding 2), a natural question arises: can we retain the flexibility of AO-GPT while mitigating its convergence drawbacks, perhaps by guiding it with some explicit L2R signal?

To explore this, we investigate the effect of incorporating a small fraction of standard Left-to-Right (L2R) ordered data directly into the training process of an AO-GPT. Specifically, we modify the sampling of generation orders $\sigma$ such that 90% of training instances use an order sampled uniformly from $S_n$ (i.e., $\sigma \sim \mathcal{U}(S_n)$), while the remaining 10% of instances use the fixed L2R order ($\sigma = \text{id}$). The impact of this hybrid approach on training dynamics and performance is illustrated in Figure 2(b). Also, as shown in Table 1 under the "Left-to-Right" evaluation, our AO-GPT model incorporating this 10% L2R data achieves significantly lower zero-shot validation perplexity compared to a purely any-order trained $\sigma$-GPT, underscoring the substantial improvement in final loss achieved by this hybrid approach. This experiment yields two significant observations:

> **Finding 3.1:** *Incorporating a small fraction (10%) of explicitly Left-to-Right ordered data into the training of an AO-GPT drastically improves its performance (both convergence speed and final loss) when evaluated on standard Left-to-Right.*

This result, while perhaps not entirely unexpected given the inherent structure of language, is starkly illustrated when comparing final performance metrics. More surprisingly, this hybrid training strategy also benefits the model's any-order capabilities:

> **Finding 3.2:** *A small fraction (10%) of Left-to-Right data can even improve the model performance on Any-Order data.*

Finding 3.2 is particularly intriguing. It suggests that highly structured L2R patterns provide a beneficial inductive bias or a more stable learning signal, helping the model learn fundamental

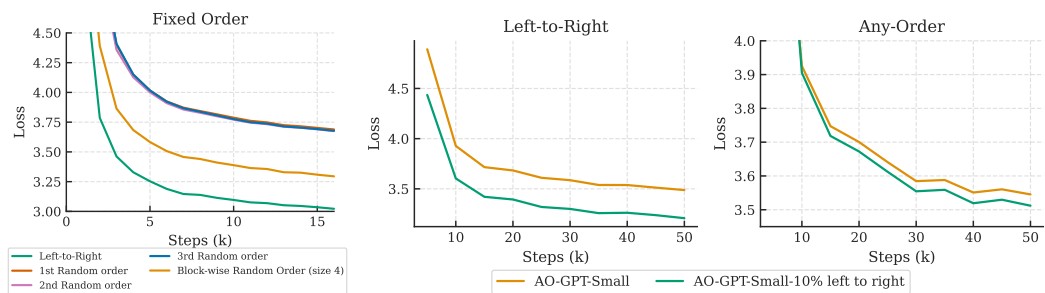

Figure 2: (a) Convergence speed with different fixed prediction orders: left-to-right, fixed random, and fixed block-wise random. (b) Impact of adding 10% left-to-right (L2R) data to AO-GPT training on its L2R and any-order loss.

linguistic structures more effectively. This, in turn, appears to enhance generalization across arbitrary permutations, rather than being a mere data trade-off. This phenomenon itself warrants dedicated investigation beyond our initial exploration, as a comprehensive understanding of this cross-order benefit is a promising direction for future work.

## 4 ENCODER-ONLY AND DECODER-ONLY MASKED DIFFUSION MODELS

Building on the insights from comparing AR and AO-AR within a decoder-only framework, we now turn our attention to the impact of the architectural choice itself when implementing an Any-Order Autoregressive (or Masked Diffusion Model) formulation. This leads to our second research question:

> **Question 2:** *For Masked Diffusion Models (or AO-AR) on language tasks, what are the theoretical and empirical differences using an encoder-only versus a decoder-only architecture?*

To address Question 2, we analyze how encoder-only and decoder-only architectures differ in modeling the univariate conditional probabilities $p(\boldsymbol{x}_j|\boldsymbol{x}_E)$ that underpin AO-AR or MDM, where $j$ is the index of the target token and $E \subset \{1, \ldots, n\} \setminus \{j\}$ is the index set of observed context tokens $\boldsymbol{x}_E$.

### 4.1 MODELING UNIVARIATE CONDITIONAL PROBABILITIES

**Order-Invariant Formulation:** For encoder-only AO-AR, the computed conditional probability $p_{\boldsymbol{\theta}}(\boldsymbol{x}_j|\boldsymbol{x}_E)$ is *order-invariant*. Provided that each token in $\boldsymbol{x}_E$ is correctly paired with its positional encoding, the permutation of these token-position pairs within the input does not alter the output probability for $\boldsymbol{x}_j$.

**Order-Dependent Formulation:** For decoder-only AO-AR, the inherent asymmetry in causal attention means the prediction $p_{\boldsymbol{\theta}}(\boldsymbol{x}_j|\boldsymbol{x}_E, \sigma_E)$ is *order-dependent* even each token in $\boldsymbol{x}_E$ is correctly paired with its positional encoding. The probability explicitly depends on the sequence $\sigma_E$ in which the context tokens $\boldsymbol{x}_E$ are presented to the model.

Henceforth, we refer to $p_{\boldsymbol{\theta}}(\boldsymbol{x}_j|\boldsymbol{x}_E)$ as an *order-invariant* conditional probability and $p_{\boldsymbol{\theta}}(\boldsymbol{x}_j|\boldsymbol{x}_E, \sigma_E)$ as an *order-dependent* conditional probability. The distinction between order-invariant and order-dependent modeling leads to different counts of the effective conditional probability space covered by each architecture.

> **Finding 5:** *Encoder-only AO-AR models $n \cdot 2^{n-1}$ order-invariant univariate conditional probability while decoder-only AO-AR models approximately $e \cdot n!$ order-dependent univariate conditional probability.*

**Encoder-only AO-AR:** This architecture models the probability of predicting any token $\boldsymbol{x}_j$ ($n$ possibilities) given any subset $\boldsymbol{x}_E$ of the other $n-1$ tokens. Since the order of $\boldsymbol{x}_E$ does not matter, we count the number of distinct pairs $(j, E)$. For each target $j$, there are $2^{n-1}$ possible subsets $E$. Therefore, the model represents $n \cdot 2^{n-1}$ unique *order-invariant* conditional probabilities. This quantity can be

Table 1: Zero-shot validation perplexity($\downarrow$) on a variety of datasets on GPT-2-medium sized models ($\sim$350M). [†]Model reproduced by the authors. [‡]Model trained with an additional 10% left-to-right ordered data.

| Model | LAMBADA | WikiText2 | PTB | WikiText103 | 1BW |
|---|---|---|---|---|---|
| *Left-to-Right* | | | | | |
| *Encoder-only Models* | | | | | |
| SEDD | 39.19 | 30.29 | 88.18 | 30.05 | 53.54 |
| RADD | **38.60** | **29.71** | **76.00** | **29.96** | **52.36** |
| *Decoder-only Models* | | | | | |
| GPT-2 | **35.66** | 31.80 | 123.14 | 31.39 | 55.72 |
| $\sigma$-GPT[†] | 61.05 | 44.80 | 121.48 | 43.68 | 76.74 |
| *AO-GPT*[‡] | 42.44 | **31.52** | **87.56** | **30.86** | **50.23** |
| *Any-Order* | | | | | |
| *Encoder-only Models* | | | | | |
| SEDD | 42.77 | 31.04 | 87.12 | 29.98 | 61.19 |
| RADD | **41.96** | **29.96** | **79.06** | **28.51** | **57.07** |
| *Decoder-only Models* | | | | | |
| $\sigma$-GPT[†] | 57.27 | 43.28 | 126.11 | 41.80 | 74.81 |
| *AO-GPT*[‡] | 49.79 | 33.73 | 101.01 | 32.38 | 65.90 |
| *AO-GPT*[‡] + *Ensembles* (8 times) | 45.08 | 30.63 | 87.74 | 29.46 | 59.11 |
| *AO-GPT*[‡] + *Ensembles* (64 times) | **44.31** | **30.16** | **85.75** | **28.98** | **58.18** |

derived combinatorially: summing over all possible context sizes $k$ (from 0 to $n-1$), we choose $k$ context tokens ($\binom{n}{k}$ ways) and have $n-k$ possible target tokens. The total is $\sum_{k=0}^{n-1} \binom{n}{k}(n-k) = n \cdot 2^{n-1}$.

**Decoder-only AO-AR:** This architecture models *order-dependent* probabilities. While it can potentially generate samples according to any of the $n!$ permutations, the fundamental units are $p_{\boldsymbol{\theta}}(\boldsymbol{x}_j|\boldsymbol{x}_E, \sigma_E)$. To count the number of distinct such terms, we again sum over context size $k$. For a fixed context set $E$ of size $k$ and a fixed target $j$, there are $k!$ possible orderings $\sigma_E$. Therefore, the total number of distinct order-dependent probabilities is:

$$N_{\text{ordered}} = \sum_{k=0}^{n-1} \underbrace{\binom{n}{k}}_{\substack{\text{Choose} \\ \text{context}}} \underbrace{(n-k)}_{\substack{\text{Choose} \\ \text{target}}} \underbrace{k!}_{\substack{\text{Order} \\ \text{context}}} = \sum_{k=0}^{n-1} \frac{n!}{k!(n-k)!}(n-k)k!$$

$$= \sum_{k=0}^{n-1} \frac{n!}{(n-k-1)!} = n! \sum_{i=0}^{n-1} \frac{1}{i!} \quad (\text{letting } i = n-k-1) \tag{6}$$

As $n \to \infty$, this sum quickly converges to $e \cdot n!$. This significantly larger number compared to the encoder-only case reflects the decoder's sensitivity to the permutation of the context. The core difference highlighted by these counts is whether the model's prediction $p(\boldsymbol{x}_j|\dots)$ is conditioned on the context as an unordered set $\boldsymbol{x}_E$ (encoder) or as an ordered sequence $(\boldsymbol{x}_{\sigma_E(1)}, \dots, \boldsymbol{x}_{\sigma_E(k)})$ (decoder).

## 4.2 ENSEMBLE ON CONTEXT ORDER

As shown in Figure 3, decoder-only AO-AR models exhibit higher zero-shot perplexity than their encoder-only counterparts. We hypothesize this performance gap is due to the inherently harder task faced by decoders: they must learn to represent approximately $e \cdot n!$ distinct order-dependent conditional probabilities, a vastly larger space than the $n \cdot 2^{n-1}$ order-invariant conditionals modeled by encoders. Given that the increased number is completely due to the order of context tokens $\boldsymbol{x}_{\sigma_{<i}}$, we introduce an ensembling technique for evaluating $\mathcal{L}_{\text{AO-AR}} = \mathbb{E}_{\sigma \sim \mathcal{U}(S_n)} \left[ \sum_{i=1}^{n} -\log p_{\boldsymbol{\theta}}(\boldsymbol{x}_{\sigma_i}|\boldsymbol{x}_{\sigma_{<i}}) \right]$. For each individual conditional probability $p_{\boldsymbol{\theta}}(\boldsymbol{x}_{\sigma_i}|\boldsymbol{x}_{\sigma_{<i}})$, we generate $M$ random permutations of the context sequence $\boldsymbol{x}_{\sigma_{<i}}$. The probability used for the loss calculation, $p_{\text{ens}}(\boldsymbol{x}_{\sigma_i}|\boldsymbol{x}_{\sigma_{<i}})$, is obtained by averaging the model's predictions across $M$ permutations of the context $\boldsymbol{x}_{\sigma_{<i}}$:

$$p_{\text{ens}}(\boldsymbol{x}_{\sigma_i}|\boldsymbol{x}_{\sigma_{<i}}) = \frac{1}{M} \sum_{j=1}^{M} p_{\boldsymbol{\theta}}\left(\boldsymbol{x}_{\sigma_i}|\boldsymbol{x}_{\text{perm}_j(\sigma_{<i})}, \text{perm}_j(\sigma_{<i})\right). \tag{7}$$

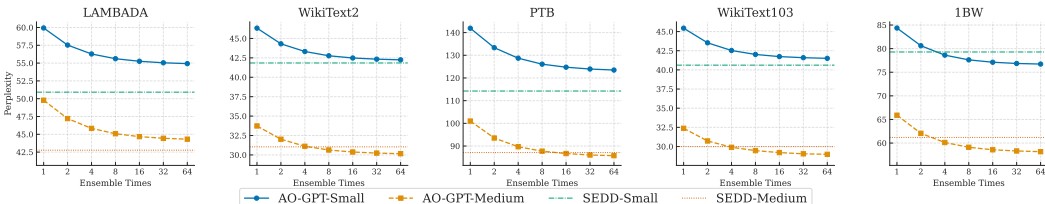

Figure 3: Zero-shot unconditional perplexity ($\downarrow$) for varying ensemble sizes. An ensemble size of 1 represents the baseline model without ensembling.

Here, $\text{perm}_j$ signifies the $j$-th permutation of the context sequence. (We always include the identity permutation among the $M$ permutations in practice.) Crucially, while the order of the input sequence fed to the model is permuted, each token remains associated with its original positional encoding. This averaging process approximates the order-invariance of an encoder by marginalizing over the context order. The results in Figure 3 show that the ensemble on order context fills the gap.

> **Finding 6:** *Decoder-only AO-AR fall shorts of their Encoder-only counterpart, while ensemble on order context fill the gap.*

### 4.3 GENERATION COMPUTATIONAL COMPLEXITY

The reverse process in MDMs iteratively recovers masked tokens as follows:

$$q_{s|t} = \prod_{i=0}^{n-1} q_{s|t}(\boldsymbol{x}_s^i|\boldsymbol{x}_t), \text{ where } q_{s|t}(\boldsymbol{x}_s^i|\boldsymbol{x}_t) = \begin{cases} 1, & \boldsymbol{x}_t^i \neq [\text{MASK}], \boldsymbol{x}_s^i = \boldsymbol{x}_t^i \\ \frac{s}{t}, & \boldsymbol{x}_t^i = [\text{MASK}], \boldsymbol{x}_s^i = [\text{MASK}] \\ \frac{t-s}{t} q_{0|t}(\boldsymbol{x}_s^i|\boldsymbol{x}_t), & \boldsymbol{x}_t^i = [\text{MASK}], \boldsymbol{x}_s^i \neq [\text{MASK}]. \end{cases}$$
(8)

Here, $q_{0|t}(\boldsymbol{x}_s^i|\boldsymbol{x}_t)$ (when $\boldsymbol{x}_t^i = [\text{MASK}]$) represents a distribution over the vocabulary for predicting a non-[MASK] token, provided by the model. Sampling $\boldsymbol{x}_s$ from $q_{s|t}(\boldsymbol{x}_s|\boldsymbol{x}_t)$ involves sampling each $\boldsymbol{x}_s^i$ independently. For positions $i$ where $\boldsymbol{x}_t^i \neq [\text{MASK}]$, $\boldsymbol{x}_s^i$ is deterministically set to $\boldsymbol{x}_t^i$. For positions where $\boldsymbol{x}_t^i = [\text{MASK}]$, a more efficient two-stage sampling procedure can be employed, as stated in Lemma 1.

**Lemma 1** (Efficient Sampling Algorithm). *For sampling $\boldsymbol{x}_s^i$ from $q_{s|t}(\boldsymbol{x}_s^i|\boldsymbol{x}_t)$ as defined in Equation (8) when $\boldsymbol{x}_t^i = [MASK]$, an equivalent sampling procedure is:*

*1. Sample a binary variable $b \sim \text{Bernoulli}\left(\frac{s}{t}\right)$.*

*2. If $b = 1$, set $\boldsymbol{x}_s^i = [MASK]$.*

*3. If $b = 0$, sample $\boldsymbol{x}_s^i$ from the distribution $q_{0|t}(\cdot|\boldsymbol{x}_t)$. It reduces computational cost by only requiring the evaluation of $q_{0|t}(\cdot|\boldsymbol{x}_t)$ when $b = 0$. The proof of equivalence and a detailed discussion of the computational benefits are provided in Appendix D.*

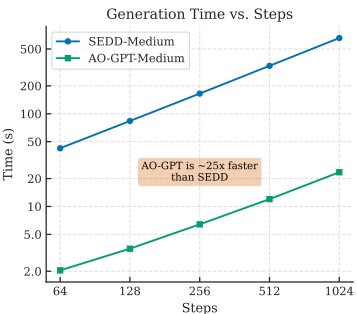

Figure 4: Generation time versus number of generation steps with sequence length $1024$ and batch size$= 32$ for decoder-only AO-AR models (with KV-cache and Lemma 1) and their encoder-only counterparts (SEDD).

Leveraging both the KV-cache mechanism and the efficient sampling technique from Lemma 1, decoder-only AO-AR models significantly reduce per-step computational costs. At each generation step, the KV-cache obviates recomputing context tokens, while efficient sampling (Lemma 1) restricts computation to only those tokens designated for unmasking. This synergy reduces the computational complexity of each step to $O(1)$, a marked advantage over their encoder-only counterparts. Our findings are summarized as follows:

Table 2: Generation Perplexity measured by GPT-2-Large of AO-GPT-Medium and SEDD-Medium across different generation steps and sampling settings (Top-p and Temperature). Lower values are better.

| Sampling Setting | | Steps | | | | |
|---|---|---|---|---|---|---|
| *Top-p Temperature* | *Model* | 64 | 128 | 256 | 512 | 1024 |
| 1.0, 1.0 | AO-GPT | 194.588 | 154.837 | 148.006 | 140.369 | 136.406 |
| | SEDD | 121.574 | 100.014 | 86.428 | 87.666 | 81.689 |
| 0.95, 0.9 | AO-GPT | 33.528 | 29.359 | 27.378 | 27.020 | 26.952 |
| | SEDD | 25.414 | 22.034 | 19.481 | 19.022 | 19.311 |
| 0.95, 0.7 | AO-GPT | 6.105 | 5.615 | 5.507 | 5.095 | 4.611 |
| | SEDD | 6.400 | 6.042 | 5.201 | 4.979 | 5.051 |

> **Finding 7:** *The total computational complexity of generating a length $n$ sequence using an encoder-only AO-AR is $O(n^2)$; with both KV-cache and Lemma 1, a decoder-only AO-AR's computational complexity is $O(n)$.*

The $O(n)$ total complexity for decoder-only AO-AR models (Finding 7) translates directly into tangible performance gains. This theoretical advantage is empirically validated by the significant generation speedups visualized in Figure 4. Beyond speed, we also evaluated the unconditional generation perplexity. To ensure a fair comparison and address the observation by (Zheng et al., 2024) that `float32` Gumbel noise (as used in SEDD (Lou et al., 2023)) can lower effective temperature, we utilized `float64`. We then compared AO-GPT and SEDD under three distinct annealing configurations: 1) no annealing (Top-p 1.0, Temperature 1.0), 2) mild annealing (Top-p 0.95, Temperature 0.9), and 3) appropriate annealing (Top-p 0.95, Temperature 0.7). The detailed results of generation perplexity are in Table 2, with the main conclusions summarized in Findings 8.1 and 8.2.

> **Finding 8.1:** *AO-GPT can achieve $25\times$ speedup on generation compared with SEDD.*
> **Finding 8.2:** *AO-GPT exhibits higher generation perplexity without logit annealing, while appropriate annealing brings its perplexity to a comparable level.*

The observed perplexity difference, particularly without annealing (Finding 8.2), can be attributed to AO-GPT's lower modeling likelihood compared to SEDD when considered in non-ensembled configurations. Thus, while SEDD might achieve better perplexity for models of comparable size, AO-GPT's striking 25x speed advantage (Finding 8.1) presents a compelling trade-off between generation quality and practical inference speed. The findings in this section illuminate the significant distinctions between encoder-centric and decoder-centric approaches, suggesting that exploring how to synergistically combine their respective advantages is a crucial direction for future work.

## 5 CONCLUSION AND LIMITATION

In this work, we conducted a systematic investigation to decouple the effects of modeling paradigms (AR vs. MDM) from their commonly associated architectural choices (decoder-only vs. encoder-only). By implementing MDMs (as AO-AR) within a decoder-only framework, we facilitated a more equitable comparison and explored architectural influences within the MDM paradigm itself.

Our experiments were conducted on models of up to medium size (e.g., 350M parameters). Whether these observations generalize to significantly larger computational scales remains an open question. Furthermore, this work focused on language; the applicability of our findings to other discrete data modalities is uncertain, especially as many such modalities may not possess the strong left-to-right sequential structure inherent in natural language.

The development of our AO-GPT model itself represents a significant contribution. To enhance any-order modeling within decoder-only architectures, we conducted extensive architectural ablations, particularly concerning the injection of target position information, and explored training strategy improvements such as Exponential Moving Average (EMA) and order-mixtures. Further details on these model-specific developments are provided in Appendix A.

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

APPENDIX

**LLM Usage**   In the preparation of this manuscript, we employed large language models to provide language-related assistance. Specifically, the LLM was used to (i) polish grammar, style, and readability of the text; (ii) offer suggestions for clearer phrasing and more concise expression.

The appendix provides supplementary material to the main paper, beginning with Section A which introduces AO-GPT, a decoder-only model aimed at unifying autoregressive and masked diffusion paradigms. This section details its broader significance, architectural considerations for injecting target position information, and an exploration of its design space, including ablation studies on target position injection strategies like adaptive LayerNorm (adaLN) and the impact of Exponential Moving Average (EMA). Following this, Section B offers further experimental specifics, including model hyperparameters and architectural details for different AO-GPT sizes, alongside additional results such as zero-shot validation perplexity scores for smaller models and qualitative unconditional text samples generated by AO-GPT Medium. Section C then situates AO-GPT within the context of existing research, covering areas like any-order density estimation, other decoder-only any-order models, and randomized image generation. Finally, Section D presents a detailed proof and discussion for Lemma 1 concerning an efficient two-stage sampling procedure.

## A   AO-GPT: A DECODER-ONLY MODEL WITH THE POTENTIAL TO UNIFY AR AND MDM

### A.1   SIGNIFICANCE AND FUTURE POTENTIAL

We highlight here its broader significance and future potential. The development of efficient and effective decoder-only MDMs like AO-GPT is crucial for several reasons:

**Broader Scope:** Firstly, the strong left-to-right sequentiality inherent in natural language, which often favors standard autoregressive (AR) models, may not be as pronounced in other discrete data modalities. For domains such as biological sequences, symbolic music, or certain structured code representations, the inherent flexibility of an any-order AR approach could be more naturally suited. Implementing such models within a decoder-only architecture, as AO-GPT proposes, could offer substantial advantages over rigidly ordered AR models, potentially leading to more effective modeling and generation in these diverse fields.

**Efficiency:** Secondly, within the realm of language modeling itself, the pursuit of decoder-only MDMs is driven by a compelling trade-off between modeling paradigms and computational efficiency. Decoder-only AO-AR offers significant theoretical and empirical advantages in density estimation and generation in terms of theoretical computational complexity and empirical generation speed (as discussed in Section 2.2, 4 and indicated by Finding 7, 8.1).

**Flexibility:** Thirdly, decoder-only AO-AR frameworks like AO-GPT exhibit superior flexibility in controlling the distribution over token generation orders. While encoder-only AO-AR models can vary the distribution of the number of masked tokens, they cannot easily specialize to a strict, fixed-order autoregressive model (e.g., left-to-right). In contrast, decoder-only AO-AR models can directly learn from any permutation distribution $P(S_n)$. This allows them to instantiate a standard left-to-right AR model by simply using the identity permutation ($\sigma = \mathrm{id}$), or to train on hybrid order distributions (as shown in Section 3). This adaptability uniquely positions them to interpolate between, and potentially unify, autoregressive and masked diffusion paradigms within a single architecture.

### A.2   INJECTING TARGET POSITION INFORMATION

In this section, we will give a comprehensive description of the modeling details, training, and inference of masked diffusion models using a decoder-only architecture. To train autoregressive models on sequences in *any order* with a decoder-only model, a key architectural modification is necessary compared to standard GPT: adding explicit *target position* information. In a traditional AR model setup (like GPT), the model implicitly predicts the token immediately succeeding the current one; the target position is always the next index in the sequence.

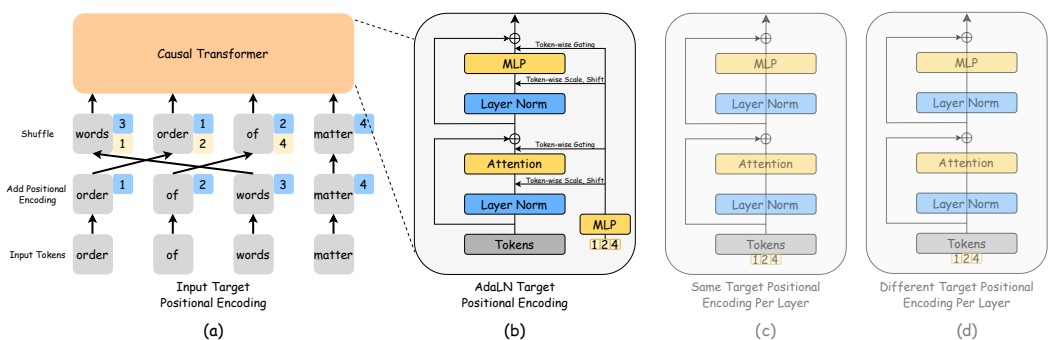

Figure 5: Target position injection strategies for decoder-only AO-AR model.

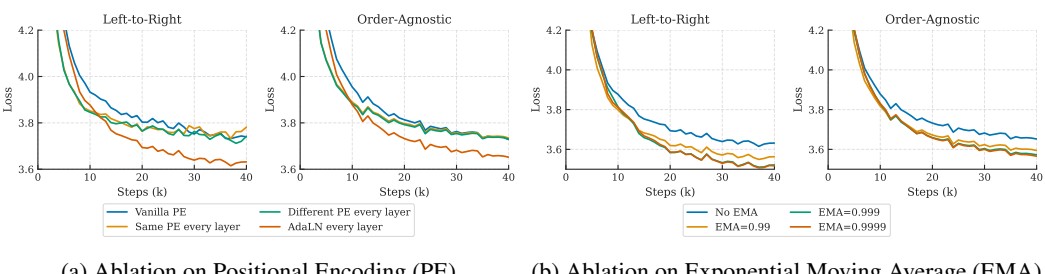

(a) Ablation on Positional Encoding (PE)          (b) Ablation on Exponential Moving Average (EMA)

Figure 6: Ablation studies for AO-GPT: target positional encoding and exponential moving average.

However, when processing sequences in a shuffled order according to a permutation $\sigma = (\sigma_1, \ldots, \sigma_n)$, the token at step $t$ in the shuffled sequence is $\boldsymbol{x}_{\sigma_t}$, and the target token to predict at step $t$ is $\boldsymbol{x}_{\sigma_{t+1}}$. The original position $\sigma_{t+1}$ is not fixed relative to the current step $t$, but varies depending on the specific permutation. Therefore, at step $t$, to predict $\boldsymbol{x}_{\sigma_{t+1}}$, the model should access the information of the input representation for tokens $\boldsymbol{x}_{\sigma_{\leq t}}$, its position in the original sequence ($\sigma_{\leq t}$), and crucially, the original position $\sigma_{t+1}$ of the next token to be predicted in the shuffled sequence. This is necessary because transformers need the explicit target original position ($\sigma_{t+1}$) to identify which specific original position's token it should predict next, a requirement absent in fixed-order prediction.

We identify two existing decoder-only architectures for training order-agnostic autoregressive models: XL-Net (Yang et al., 2019) and $\sigma$-GPT (Pannatier et al., 2024). XL-Net incorporates the target position using two-stream attention, a mechanism that differs significantly from mainstream decoder architectures. Therefore, we do not adopt this approach. In contrast, $\sigma$-GPT incorporates the target position through an additional target positional encoding on a standard GPT architecture as Figure 5(a). Thus, we choose to adopt $\sigma$-GPT as a baseline method.

### A.3 DESIGN SPACE OF AO-GPT

#### A.3.1 TARGET POSITION INJECTIONS

We observe slow convergence in $\sigma$-GPT, particularly during its initial training stages. Beyond the factors analyzed in Section 3, we hypothesize that the semantic requirements imposed by the target position significantly influence the predicted token, potentially more so than a simple target position encoding can adequately capture. For instance, in the sentence `She put the _ in the _`, the first blank typically requires a noun representing a movable object (e.g., `book`, `key`, `food`), while the second necessitates a noun denoting a container or location (e.g., `box`, `drawer`, `fridge`). The distinct lexical distributions for these two positions suggest that a single, undifferentiated target position encoding applied after the token embedding may be insufficient to model these nuanced, position-dependent semantic constraints.

To address this potential limitation, we propose and ablate three distinct strategies for incorporating richer target position information, all designed to incur negligible additional computational cost:

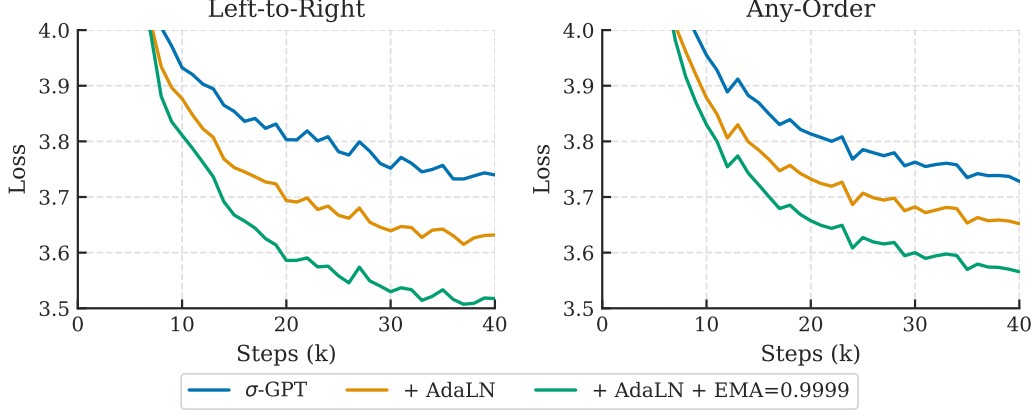

Figure 7: Combined impact of adaptive layerNorm (AdaLN) and exponential moving average (EMA) on AO-GPT training. Loss curves compare the baseline $\sigma$-GPT, $\sigma$-GPT with AdaLN, and $\sigma$-GPT with both AdaLN and EMA (decay 0.9999) for Left-to-Right (left) and Any-Order (right) objectives.

Table 3: Zero-shot validation perplexity($\downarrow$) on a variety of datasets on GPT-2-medium sized models ($\sim$350M). [†]Model reproduced by the authors. [‡]Model trained with an additional 10% left-to-right ordered data.

| Model | LAMBADA | WikiText2 | PTB | WikiText103 | 1BW |
|---|---|---|---|---|---|
| *Left-to-Right* | | | | | |
| $\sigma$-GPT[†] | 61.05 | 44.80 | 121.48 | 43.68 | 76.74 |
| *AO-GPT*[‡] | **42.44** | **31.52** | **87.56** | **30.86** | **50.23** |
| *Any-Order* | | | | | |
| $\sigma$-GPT[†] | 57.27 | 43.28 | 126.11 | 41.80 | 74.81 |
| *AO-GPT*[‡] | **49.79** | **33.73** | **101.01** | **32.38** | **65.90** |

(1) Figure 5(c) re-applying the same target positional encoding at the input of each Transformer block; (2) Figure 5(d) utilizing distinct, learnable target positional encodings for each Transformer block; and (3) Figure 5(b) conditioning the LayerNorm parameters (Perez et al., 2018; Peebles & Xie, 2023) within each Transformer block on the target position. As shown in Figure 6, the former two ways demonstrate some early training acceleration compared to the baseline, but their advantages diminish in later stages, eventually performing nearly identically to the baseline. In contrast, the adaptive LayerNorm (adaLN) approach shows consistent improvements throughout the entire training process. This suggests that dynamically modulating LayerNorm parameters based on target position provides a more effective and stable way to incorporate positional information, likely because it allows for finer-grained, context-dependent normalization at each transformer block.

### A.3.2    EXPONENTIAL MOVING AVERAGE

While Exponential Moving Average (EMA) of model weights is a less common technique in the pre-training of standard autoregressive language models, it is a widely adopted practice in the training of both continuous and current discrete diffusion models, where it often contributes to improved sample quality and training stability. Given the potential for EMA to smooth the training trajectory, and potentially to help smooth out noise, allowing the optimization to converge to the target loss more efficiently, we conducted an ablation study to assess its impact on OA-GPT. We experimented with several EMA decay rates: 0.99, 0.999, and 0.9999, comparing these against a baseline model trained without EMA. Our results indicate a clear benefit to employing EMA. All tested EMA configurations outperformed the baseline model (no EMA). Notably, an EMA decay rate of 0.9999 yielded the best performance among the values tested.

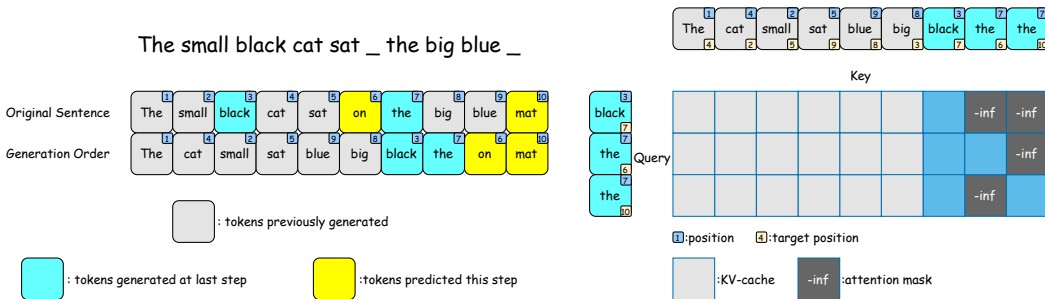

Figure 8: Attention mask for simultaneous prediction of multiple tokens in AO-GPT.

Having established the individual benefits of adaptive layerNorm (adaLN) for target position injection and exponential moving average (EMA) for training, we investigate their combined effect on AO-GPT. Figure 7 illustrates the training loss progression for both Left-to-Right and Any-Order objectives when integrating both adaLN and EMA (with a decay of 0.9999) into the $\sigma$-GPT baseline. The results demonstrate that these two enhancements are largely orthogonal, with their combination leading to significantly improved convergence and lower final loss values compared to the baseline $\sigma$-GPT and models with only one of the improvements. This synergistic effect is further corroborated by the zero-shot perplexity scores presented in Table 3, where the fully enhanced AO-GPT (incorporating adaLN, EMA, and 10% L2R data) substantially outperforms the reproduced $\sigma$-GPT baseline across all evaluated datasets for both Left-to-Right and Any-Order evaluations.

### A.4 PARALLEL GENERATION ATTENTION MASK

The reverse process in MDMs iteratively recovers masked tokens as follows:

$$q_{s|t} = \prod_{i=0}^{n-1} q_{s|t}(\boldsymbol{x}_s^i|\boldsymbol{x}_t), \text{where } q_{s|t}(\boldsymbol{x}_s^i|\boldsymbol{x}_t) = \begin{cases} 1, & \boldsymbol{x}_t^i \neq [\text{MASK}], \boldsymbol{x}_s^i = \boldsymbol{x}_t^i \\ \frac{s}{t}, & \boldsymbol{x}_t^i = [\text{MASK}], \boldsymbol{x}_s^i = [\text{MASK}] \\ \frac{t-s}{t} q_{0|t}(\boldsymbol{x}_s^i|\boldsymbol{x}_t), & \boldsymbol{x}_t^i = [\text{MASK}], \boldsymbol{x}_s^i \neq [\text{MASK}]. \end{cases}$$

$$(9)$$

Here, $q_{0|t}(\boldsymbol{x}_t^i|\boldsymbol{x}_t)$ (when $\boldsymbol{x}_t^i = [\text{MASK}]$) represents a distribution over the vocabulary for predicting a non-[MASK] token, provided by the model. Sampling $\boldsymbol{x}_s$ from $q_{s|t}(\boldsymbol{x}_s|\boldsymbol{x}_t)$ involves sampling each $\boldsymbol{x}_s^i$ independently. Crucially, AO-GPT, equipped with the specialized attention mask depicted in Figure 8, can compute these probabilities $q_{0|t}(\boldsymbol{x}_s^i|\boldsymbol{x}_t)$ for different $i$ in a single forward pass. This mask ensures that the prediction for each masked token $\boldsymbol{x}_s^i$ is conditioned only on the unmasked tokens present in $\boldsymbol{x}_t$ and its own position, without attending to other concurrently predicted masked tokens. Consequently, this parallel prediction scheme for multiple masked tokens introduces no training/inference mismatch for the individual $q_{0|t}(\boldsymbol{x}_s^i|\boldsymbol{x}_t)$ distributions, as each is generated under conditions consistent with the model's training.

## B ADDITIONAL EXPERIMENT DETAILS AND RESULTS

### B.1 MODEL DETAILS

The AO-GPT models were trained with several common hyperparameters. Specifically, both Small and Medium models used a $d_{\text{head}}$ of 64, a batch size of 0.5M tokens, a weight decay of 0.05, Adam optimizer parameters $\beta_1 = 0.9$ and $\beta_2 = 0.95$, and an Exponential Moving Average (EMA) decay of 0.9999. The learning rate was $6.0 \times 10^{-4}$ for the Small model and $3.0 \times 10^{-4}$ for the Medium model. Architectural details like $n_{\text{layers}}$, $d_{\text{model}}$, and $n_{\text{heads}}$ are the same with the GPT-2 specific to each model size as detailed in Table 4. We trained AO-GPT on nodes of 8 H800 80GB. For the input adaptive layer norm, we use a target positional encoding of 128 hidden dimensions to minimize its impact on increased parameters.

Table 4: AO-GPT Model Specifications

| Parameter | Small | Medium |
|---|---|---|
| $n_{\text{layers}}$ | 12 | 24 |
| $d_{\text{model}}$ | 768 | 1024 |
| $n_{\text{heads}}$ | 12 | 16 |
| $d_{\text{head}}$ | 64 | 64 |
| Batch Size (tokens) | 0.5M | 0.5M |
| Learning Rate | $6.0 \times 10^{-4}$ | $3.0 \times 10^{-4}$ |
| Weight Decay | 0.05 | 0.05 |
| Adam $\beta_1$ | 0.9 | 0.9 |
| Adam $\beta_2$ | 0.95 | 0.95 |
| EMA | 0.9999 | 0.9999 |

Table 5: Zero-shot validation perplexity($\downarrow$) on a variety of datasets on GPT-2-small sized models ($\sim$125M). [†]Model reproduced in this work. [‡]Model trained with an additional 10% left-to-right ordered data.

| Model | LAMBADA | WikiText2 | PTB | WikiText103 | 1BW |
|---|---|---|---|---|---|
| *Left-to-right* | | | | | |
|   *Encoder-only Models* | | | | | |
|     SEDD | 49.41 | 41.19 | 118.74 | 41.70 | 72.60 |
|     RADD | **49.09** | **38.26** | **107.78** | **38.41** | **63.33** |
|   *Decoder-only Models* | | | | | |
|     GPT-2 | **45.04** | 42.43 | 138.43 | 41.60 | 75.20 |
|     $\sigma$-GPT[†] | 68.61 | 57.66 | 146.87 | 55.54 | 90.98 |
|     *AO-GPT*[‡] | 52.46 | **42.10** | 135.96 | **40.97** | **71.73** |
| *Order-agnostic* | | | | | |
|   *Encoder-only Models* | | | | | |
|     SEDD | 50.92 | 41.84 | 114.24 | 40.62 | 79.29 |
|     RADD | **50.27** | **38.26** | **110.38** | **35.90** | **74.28** |
|   *Decoder-only Models* | | | | | |
|     $\sigma$-GPT[†] | 65.83 | 53.08 | 138.61 | 50.75 | 87.71 |
|     *AO-GPT*[‡] | 59.93 | 46.33 | 141.92 | 45.44 | 84.36 |
|     *AO-GPT*[‡] + *Ensembles* (8 times) | 55.62 | 42.77 | 126.08 | 42.02 | 77.62 |
|     *AO-GPT*[‡] + *Ensembles* (64 times) | **54.92** | **42.24** | **123.49** | **41.51** | **76.73** |

## B.2 ADDITIONAL RESULTS

Table 5 presents zero-shot validation perplexity scores on the same suite of datasets, but for models of a smaller scale (GPT-2-small size, $\sim$125M parameters). This allows for an examination of how the different approaches (SEDD, RADD, GPT-2, $\sigma$-GPT, and our AO-GPT with its enhancements) compare at different model sizes. Furthermore, to offer a qualitative assessment of AO-GPT's generative capabilities, we provide unconditional text samples generated by our AO-GPT Medium model. Figures 9, 10, and 11 showcase these generated passages under different sampling configurations (varying top-p and temperature settings). These examples illustrate the model's ability to produce coherent and contextually relevant text.

## C RELATED WORK AND BROADER IMPACT

### C.1 RELATED WORK

**Any-Order Density Estimation** Neural Autoregressive Distribution Estimation (NADE) (Uria et al., 2016) initially leveraged implicit position awareness within its MLP architecture, parameterizing conditionals $p(\boldsymbol{x}_{o_d}|p(\boldsymbol{x}_{o<d})$ for an arbitrary ordering $o$ using a weight-sharing scheme inspired by RBMs. Masked Autoencoder for Distribution Estimation (MADE) (Germain et al., 2015) adapted standard

autoencoders by carefully masking connections to enforce autoregressive properties for arbitrary orders. Autoregressive Diffusion Models (ARDMs) (Hoogeboom et al., 2021) integrated principles from diffusion processes, training a shared network on masked modeling objective. Arbitrary Conditional Distributions with Energy (ACE) (Strauss & Oliva, 2021) utilizes energy-based models to estimate arbitrary conditionals $p(\boldsymbol{x}_i|\boldsymbol{x}_S)$. Training and inference on any-order autoregressive models the right way (Shih et al., 2022) addresses model redundancy and training inefficiency. They propose training on a smaller set of univariate conditionals $p(\boldsymbol{x}_i|\boldsymbol{x}_S)$ and upweighting the training loss for conditionals expected to be frequent during inference, leading to improved likelihoods without sacrificing tractable inference for arbitrary conditional queries. InDIGO (Gu et al., 2019) introduces a novel insertion-based decoding algorithm for Transformers, enabling flexible sequence generation in arbitrary orders. While InDIGO demonstrates adaptive generation, our work further investigates the implications of such order-agnostic capabilities within the discrete diffusion framework and its architectural consequences for likelihood modeling.

**Existing Decoder-only Any-Order Model** Adapting decoder-only Transformers for any-order tasks has led to distinct approaches. XLNet (Yang et al., 2019) trained the model to predict tokens in all possible factorization orders to capture bidirectional context. It incorporates the target position $z_t$ using a two-stream self-attention mechanism (content stream $h_{z_t}$ and query stream $g_{z_t}$), a mechanism that differs significantly from mainstream decoder architectures. $\sigma$-GPT (Pannatier et al., 2024) enables any-order generation in a standard GPT architecture by incorporating the target position through an additional concatenated target positional encoding. Specifically, to predict token $\boldsymbol{x}_{\sigma(t+1)}$, the model receives the current token $\boldsymbol{x}_{\sigma(t)}$, its original position $\sigma(t)$, and the original position of the target token $\sigma(t+1)$, allowing any generation order.

**Randomized Image Generation** Randomized Autoregressive modeling (RAR) by Yu et al. followed $\sigma$-GPT (Pannatier et al., 2024) and trained a standard autoregressive model with an additional target position encoding on permuted image tokens with annealing during training. Rand-AR, proposed by Pang et al., is a decoder-only visual model trained on randomly permuted image tokens, using an explicit position instruction token (which doubles the token length) before each image token to specify the spatial location of the next token to be predicted. While RAR and Rand-AR both explored training on permuted sequences, they primarily focused on the visual domain and differ from our work in several key aspects. **(1)** these methods operate exclusively on image tokens. The statistical properties and inherent structures of image tokens (e.g., local spatial correlations) are substantially different from those of language tokens, which exhibit more complex, long-range semantic and grammatical dependencies. **(2)** their approach to generation and the theoretical framework varies. RAR, despite training with permutations, ultimately anneals towards and generates images using a conventional raster scan order. Rand-AR does consider parallel decoding by predicting tokens for specified positions; however, it does not explicitly investigate or establish the connection between its order-agnostic generation and the principles of discrete diffusion models, a central aspect of our study. **(3)** the evaluation focus of these vision-centric works is predominantly on generative quality metrics such as Fréchet Inception Distance (FID) and Inception Score (IS). In contrast, our research places a strong emphasis on understanding the fundamental differences in likelihood modeling capabilities (e.g., perplexity) between standard autoregressive and order-agnostic/masked diffusion paradigms, particularly when controlling for architectural choices.

## C.2 BROADER IMPACT

This research contributes to the development of text generation models with potentially enhanced controllability. This work explores diverse pathways for language modeling without inherently aiming to escalate the raw generative power beyond current systems, thereby promoting responsible exploration of AI techniques.

## D PROOF AND DISCUSSION FOR LEMMA 1

Lemma 1 states that for sampling $\boldsymbol{x}_s^i$ from $q_{s|t}(\boldsymbol{x}_s^i|\boldsymbol{x}_t)$ (Equation (8) in the main text) when $\boldsymbol{x}_t^i = $ [MASK], an equivalent two-stage sampling procedure can be used which reduces computational cost. We provide the proof of equivalence and discuss the computational advantages here.

*Proof.* When $\boldsymbol{x}_t^i = $ [MASK], the original distribution $q_{s|t}(\boldsymbol{x}_s^i|\boldsymbol{x}_t)$ is defined as:

- $P(\boldsymbol{x}_s^i = [\text{MASK}]|\boldsymbol{x}_t) = \frac{s}{t}$

- $P(\boldsymbol{x}_s^i = v|\boldsymbol{x}_t) = \frac{t-s}{t} q_{0|t}(v|\boldsymbol{x}_t)$, for any token $v \neq [\text{MASK}]$.

Here $q_{0|t}(v|\boldsymbol{x}_t)$ is the probability of token $v$ given by the model's predictive distribution for the masked position. The proposed procedure for $\boldsymbol{x}_t^i = [\text{MASK}]$ is:

**Stage 1: Bernoulli Trial.** Sample a binary variable $b \sim \text{Bernoulli}\left(\frac{s}{t}\right)$.

**Stage 2: Determine $\boldsymbol{x}_s^i$.**

- If $b = 1$ (occurs with probability $\frac{s}{t}$), set $\boldsymbol{x}_s^i = [\text{MASK}]$.

- If $b = 0$ (occurs with probability $1 - \frac{s}{t} = \frac{t-s}{t}$), then sample $\boldsymbol{x}_s^i$ from the distribution $q_{0|t}(\cdot|\boldsymbol{x}_t)$.

We demonstrate that this two-stage procedure generates samples with probabilities identical to the original definition of $q_{s|t}(\boldsymbol{x}_s^i|\boldsymbol{x}_t)$ when $\boldsymbol{x}_t^i = [\text{MASK}]$.

**Probability of sampling $\boldsymbol{x}_s^i = [\text{MASK}]$ with the new procedure**: This event occurs if and only if the Bernoulli trial in Stage 1 yields $b = 1$. $P_{\text{new}}(\boldsymbol{x}_s^i = [\text{MASK}]|\boldsymbol{x}_t) = P(b = 1) = \frac{s}{t}$. This matches the probability $P(\boldsymbol{x}_s^i = [\text{MASK}]|\boldsymbol{x}_t)$ from the original definition.

**Probability of sampling $\boldsymbol{x}_s^i = v$ (where $v \neq [\text{MASK}]$) with the new procedure**: This event occurs if and only if the Bernoulli trial in Stage 1 yields $b = 0$, AND $\boldsymbol{x}_s^i$ is subsequently sampled as $v$ from $q_{0|t}(\cdot|\boldsymbol{x}_t)$ in Stage 2. The probability is:

$$
\begin{aligned}
P_{\text{new}}(\boldsymbol{x}_s^i = v|\boldsymbol{x}_t) &= P(b = 0 \text{ and } \boldsymbol{x}_s^i \text{ is sampled as } v \text{ from } q_{0|t}) \\
&= P(b = 0) \times P(\text{sample } v \text{ from } q_{0|t}|b = 0) \\
&= \left(1 - \frac{s}{t}\right) \times q_{0|t}(v|\boldsymbol{x}_t) \\
&= \frac{t-s}{t} q_{0|t}(v|\boldsymbol{x}_t)
\end{aligned}
$$

This also matches the probability $P(\boldsymbol{x}_s^i = v|\boldsymbol{x}_t)$ from the original definition for $v \neq [\text{MASK}]$. Since the probabilities for all possible outcomes of $\boldsymbol{x}_s^i$ (either $[\text{MASK}]$ or any $v \neq [\text{MASK}]$) are identical under both the original definition and the two-stage sampling procedure, the two methods are mathematically equivalent. $\qquad \square$

. That's part of it. That's what their challenges are.

Price cuts kept going up, and regressive tax cuts used to avoid the biggest recession deficits ever.

And gave Bill Clinton earlier everything that Frederick Douglass said would, and will, help provide stimulus, public goods and extraordinary discovery of opposition to Republican New Deal. Like Dewey, failed Eisenhower, Bunyan, and other story, Democratic nomination does not. I no need any of these huge votes. They don't have nothing, because they need a party to pull.

So this is going to be difficult. You can't do more than either party got on. They can come up with a lot better than we do. They could say oboe and no sir, like Democrats do. You could say they're going to do. We frankly don't care about whether or not to. You know Eisenhower was a Republican and not a Democrat.

Hoith in the mid-1960s, in times of consensus about present threat to the budget and health care for seniors, the Democrats were real conservatives. He more than vetoed the largest military budget everyone has now. Eisenhower, did. And that continues to stand to haunt politicians. It's a third year defense, and Democrats should be the best defending it.

Still, his big difference with government-paid premiums and deductibles and the federal commitment to keep bills slightly lower, to the point.

Now Change did a little older, a lot less of aigg, delivering a public option to pay by premiums. Under his plan, seniors can subsidize your health care as it is and they won't give you the vouchers, anything. But they would let you Congress forward using the old patterns, looking at Medicare, and making sharp adjustments.

That is what we did and Eisenhower also do, and Democrats well do have an alternative to it, it's Romney. The Health Care Act, from Clinton, provided this but, you know, Democrats have opposed it, undertaking the Social Security program until roughly three years later, not after really the final reforms made Obamacare. That leaves Medicaid completely, but not the Bill Health Insurance or WILI.

Instead, these taxes will be chained for a few years. Then die, while Federal reverses itself on waiting to stop the stimulus, even now there is evidence that the American recovery, even now, has at best reached great stimulus, never zero.

Oh, you just put in a few fantasy. I have yet to run the CBO score, but it's going to keep providing two to three hundred million people, get as Census4–2008, or the amount of people who advocated to destroy those benefits.

But getting rid of the lion's taxes doesn't mean that Republican success, or it is. And there are other options. Republican care plans want to make sure the elderly and the poor don't pay taxes.

JOINS GROUP, JIM LEE, TOM MONLE, AUSTIN ZIENCEIN,

HEALTHY POLICE WOLFNEY:

SIGNED RESEARCHISM CLASS, PRINCESSES VECTOR:

—— "Intellect would rather have this conversation, my kids' children, and they're going to say how I got insurance and Medicare and we are gonna make the individuals younger, I'm saying Obama's going to get out of the wages of most Americans. But here's what my point of view is: It's constitutional. No, this is a government-run health care that is our entitlement that everybody got to raise the children of their parents." It should be a state-run entitlement.

But there's a few points Republicans should pick up on.

To some people, this really is an antitrust problem. He had a monopoly right.

And this is one issue that Republican health reform is addressing. When you think of Obamacare, one of those things is health care clearly.

Now, the antitrust of American health care is the antitrust issue.

So, to us people speak to them even if it was no problem. But the plan that would win the antitrust issue would not be Obamacare.

Kaplan Unio tried to allow health care firms to discriminate between insurers and consumers and charge them the best prices they can. With the nets rolled, the conservative media called any plan that took money to allow this micromanagement essentially pushed doctors from, essentially, price schemes.

Counter-contradictory proposal for natural-buy new surcharge. Republicans argued that people could buy insurance after union drug provisions kind of with union money.

Health insurance companies designed then still in buy in have entered discussion even saying they wouldn't have to cover abortion care. Because it's in a doctor's facility, it is a product of Blackburn's bill... It's an association. This was the bill at the highest point. Democrats rejected. They were 96% against it.... What we wanted them to say was that

Figure 9: Unconditional generation result of OA-GPT Medium (top-p= 1.0, temperature= 1.0).

that young people who were vulnerable need to be protected," the lawsuit said. "He wanted the school to be a model for people who violate our trust."

The sexual assault trial had started in January of 2014. At 17, Madsen was a popular teacher of boys at the boys' academy, but resigned abruptly in 2010. He moved into substitute teaching and working on the force. A jury in 2013 convicted him of all 49 counts of sexual abuse for allegedly performing oral sex at the boys' while he was a lieutenant.

In a similar case last year, a jury found that the district attorney's office covered up allegations of sexual misconduct that involves both female students and male students.

"It's been a legacy of defense and secrecy being preserved for many months," Barbara Cappadza, the chief deputy prosecutor for the Sacramento County District Attorney's Office, said on Thursday.

[California bureaucrat signs off deal to avoid criminal charges for covering up child sex abuse after school department violence]

"I was shocked, the shocking thing about the case, is that it seemed so away from the role she took in knowing the officers were involved in the case," she said. "They said they did have a sense of familiarity with where the case was going."

The lawsuit is the first time the office has pursued civil criminal charges, it says, against Madsen, the teacher, about the abuse. This lets Cappadza, known by the Sacramento DA as a "proactive prosecutor, prosecute all alleged cases of sexual misconduct cases.

" Complaints and actions against child sex abuse are common in all investigations," Cappadza said. "It's possible all of the criminal cases in this case go back to 2010."

In an interview, an attorney for the county attorney's office and the San Francisco Department of Children and Families, said the agency had not filed criminal charges against the current officers in the case despite the fact they did not. They have not publicly admitted to any sexual misconduct, nor that any such allegations have been made in public.

"The district attorney's office still has the authority to resolve these allegations and has determined each individual involved is in a confidential and inadvertent matter in the district attorney," Cappadza said.

The lawsuit found that the county police have brought sexual misconduct allegations against the officers about 30 times. In those cases, prosecutors find there is enough evidence even against officers and they may not be brought behind bars.

Charges against those who are fired are often steep and unusual. Madsen was previously fired from leading a school force of about 100 officers, according to a report filed by a police inspector in 2013 by the school district's police union.

Madsen complained that some of his officers were caught up in sexual misconduct accusations against officers around the country at this time over the years.

In his conference remarks, he said that there are "a growing number" of 962 sworn officers and more than 600 in 13 of California's 32 cities at large.

Highprofile cases of alleged civil rights violations in California. Take a close look into the cover-up in the 2016 rape of 11. (The New York Post)

The identity of the alleged victims could not be publicly released for the record, because prosecutors do not currently have a Madsen witness in the case.

Nonetheless, they have said that they may themselves have been victims of sexual misconduct. They would concede, however, that the allegations have been or were likely made before this case.

Human rights advocates say officers are hired and sanctioned by the public after lineups of the force.

The American Society is investigating similar complaints against a longtime member of the county police department alleging that he's disciplined twice in 2010 and 2011. Another officer reached just a guilty plea for reporting having sex with an underage student in a drug program in 2013; he then was fired in 2013 and then was fired in the same year for another job.

The suit is an unusual one against Madsen, as well as a handful of other sexual misconduct suits.

Madsen, 47, was promoted early in his tenure as the Sacramento County High School Task Force head on Jan. 21, 2010, the news outlet previously reported. He already completed three years probation after he pleaded no contest in 2012 to sexually assaulting a female student.

"I went through a lot of changes. I wanted to start it, I wanted to know about it," said Matilda Montoya, 35, who lost her sister in Honduras in the West Hills town of Telarrugada, Colombia soon after in March.

"I really wanted to get started. If I could get involved, but I got stuck here. I should get back to my trip

Figure 10: Unconditional generation result of OA-GPT Medium (top-p= 0.95, temperature= 0.9).

just say, "I'm gonna tell you, I don't put my name on the show, but I want to tell you what I think. And it's not what I want to say. And I'm gonna tell the rest of you, in my campaign and, frankly, this country. And I know the people, as many people as there are, they're going to put you on it. Because you know, you can say on the radio, "I don't care what you think. I don't want to be afraid of you. It's a shame for you."

Q: Well, it's a great shame for you.

Q: Thank you. John.

Trump: Like I said. Q: Thank you.

Trump: Thank you.

Trump: Thank you. Well, we don't have to go on and on.

Q: In the end, do you think we're going to start to hear from listening to you.

I'm going to be afraid that we're not going to be listening to you.

Trump: I think, I think, I think, that listening to the rest of the world, to the Europeans, is a danger to the world.

On the other hand, I think it's the danger, maybe, or even the peril, that the U.S. is being, right now. I think, of course, is that it has been said that we are not listening to the part of the world right now, the 1 percent of us. And the percent of people that make up 1 percent of the world, the 1 percent of Americans, and frankly, the Europeans, and the rest of the world, the rest of the rest of the world, I think that we do have the luxury of listening to the American people, in a way, listening to the Europeans, and to start listening to the world and in some ways, start to think, and think about what we're going to do to the rest of the world.

I think the United States is going to be a very powerful country, a powerful country, a very powerful military, probably the strongest military in the world, and a great power. I think that, at the end of the day, as the world's largest economy, we'll be known as a superpower. But our country is the superpower of today, and if we lose that, we're not going to be the dominant power of the area, of the world, and we're gonna be a threat to the world. We'll be a threat to Europe, we'll be in a threat to each other, and a major threat to each other. We'll be a threat, we're a threat to the other parts of the world, as the Middle East, as well, and as the Pacific, and obviously, the security world.

And that's right, but we're gonna leave, because the United States is still on the world. And if we leave, the U.S. will be the threat to the 1 percent of the world, and of the world, of the European Union, the rest of the world, and frankly, the rest of the Western world. It's a big danger for us. And we have to start to think, as Americans and the way we think about it, understand that this is important to us, because that's the relationship this is that we're going to have. We've got a great relationship with the U.S. — and we're in that relationship. It's a vital relationship.

I think it's important, I think I have to say, this don't think is necessarily, it's gonna be a loss of this, but it's gonna be more than that.

It's not be like it used to be, the rest of the world, in terms of security, we're gonna have to be smarter, I mean, and I think that we're gonna have to understand that we're in more of a danger now, as a country, than we were, than we were in the past, because we've got a system, it's going to be more complicated, we're going to have more ways to work with people all around the world, we're going to be a kind of danger.

In that sense, now I think we're in a much more danger, actually. I think this is a danger, actually, to the American public, is that much more serious and more dangerous.

That'

Figure 11: Unconditional generation result of OA-GPT Medium (top-p= 0.95, temperature= 0.7).

