# OpenReview forum: "Any-Order GPT as Masked Diffusion Model: Decoupling Formulation and Architecture"
_ICLR.cc/2026/Conference — Submitted to ICLR 2026_

### Official Review · Reviewer_rwo7 · 2025-11-01

**Soundness:** 2
**Presentation:** 3
**Contribution:** 2
**Rating:** 4
**Confidence:** 4

**Summary:**

This paper aims to fairly compare autoregressive (AR) and masked diffusion (MDM) paradigms by decoupling formulation from architecture. Prior works conflated these by pairing AR with decoder-only causal attention and MDM with encoder-only full attention.

The authors introduce AO-GPT, a decoder-only masked diffusion model that implements the Any-Order Autoregressive (AO-AR) objective, equivalent to MDM but averaged over all token permutations. Using identical architectures, they reveal:

* AO-GPT trains slower than left-to-right GPT because many random token orders are uninformative.

* Injecting 10% left-to-right data sharply improves both convergence and perplexity, showing language’s inherent sequential bias.

Compared with encoder-only MDMs, decoder-based AO-GPT models a vastly larger conditional space (~ e · n!) but achieves ≈ 25× faster generation with KV-cache and efficient sampling, approaching MDM perplexity after temperature annealing. Architecturally, AO-GPT employs adaptive LayerNorm and EMA to stabilize training, achieving near-encoder-level performance while preserving decoder efficiency.

**Strengths:**

1.  The work’s exploratory framing, decoupling modeling formulation (AR vs. MDM) from architectural choice (encoder vs. decoder), is conceptually fresh and helps clarify long-standing confusions in diffusion-based language modeling. This perspective provides some insights into how generation order and attention structure affect learning and efficiency.

2. Although limited in scale, the experiments are carefully controlled and serve as useful ablations for understanding the impact of causal vs. full attention and decoder vs. encoder designs. The results are interpretable with clear reasoning, making them valuable for guiding future model design. The finding that partial left-to-right ordering stabilizes any-order training offers a practical and actionable insight for improving masked diffusion LMs without heavy architectural change, echoing ideas from hybrid AR–diffusion works like block diffusion.

**Weaknesses:**

1. Lack of clear narrative and focus. The paper presents over eight separate findings spanning AR vs. AO-AR, encoder vs. decoder MDM, training dynamics, architectural ablations, and efficiency. This breadth makes the work feel more like a collection of exploratory observations than a cohesive study with a central takeaway. The main conceptual message, how “any-order” decoding interacts with causal attention, gets diluted by numerous side analyses and minor findings. The authors should consolidate around one or two key insights to strengthen the paper’s focus.

2. Questionable fairness premise. The notion of “fair comparison” by forcing MDM into a decoder-only causal-attention setup is debatable. Changing an encoder-based model to match the AR architecture inherently biases the comparison toward AR-style inductive biases. A symmetric control, e.g., adapting AR to an encoder/full-attention variant, would make the fairness claim more credible. As written, the work could be seen as optimizing one paradigm (MDM) to fit another’s design, rather than evaluating them on equal footing.

3. Limited novelty in methodology. The technical modifications (decoder reimplementation, order ensembling, adaptive LayerNorm, EMA) are individually incremental and borrowed from prior work (e.g., σ-GPT, diffusion model training). The novelty lies mainly in the analysis framework, but this could be emphasized more explicitly rather than presented as model contributions.

4. Small-scale experiments and weak generalization claims. All experiments are conducted at ≤350M parameters. Given the known scale sensitivity of LLM training, these results may not extrapolate to the multi-billion-parameter regime. The claimed speed–quality trade-offs, convergence behaviors, and “10% L2R” benefit might differ under large-scale or instruction-tuned settings. The paper would benefit from at least partial large-scale validation or sensitivity analysis to hyperparameters.

5. Underdeveloped connection to prior hybrid approaches. The observation that "partial autoregressive ordering" stabilizes MDM training aligns with prior “block diffusion” or “hybrid AR–diffusion” work, but this link is not acknowledged.

**Questions:**

The fairness claim is very questionable, converting MDMs into a decoder-only setup inherently imposes AR-style inductive biases, so it’s unclear why this one-directional adaptation is considered “fair,” or how conclusions might change if AR models were instead adapted to an encoder/full-attention configuration.

---

> ### Author Response · Authors · 2025-11-25
>
> >(Weakness 1) Lack of clear narrative and focus. The paper presents over eight separate findings spanning AR vs. AO-AR, encoder vs. decoder MDM, training dynamics, architectural ablations, and efficiency. This breadth makes the work feel more like a collection of exploratory observations than a cohesive study with a central takeaway. The main conceptual message, how “any-order” decoding interacts with causal attention, gets diluted by numerous side analyses and minor findings. The authors should consolidate around one or two key insights to strengthen the paper’s focus.
>
> **A:** We appreciate this feedback. The breadth of our investigation was intentional, aiming to provide a **comprehensive foundational study** of a new modeling setup (Decoder-only MDM) rather than a narrow observation. We agree, however, that the narrative focus can be sharpened. We will revise the manuscript to strictly subordinate the auxiliary findings (ablations, training dynamics) to one central thesis: **characterizing the distinct interaction between Any-Order decoding and Causal Attention.** Specifically, the paper will be framed as a systematic proof that while this interaction introduces optimization challenges (requiring specific adaptations), it fundamentally unlocks an **efficiency frontier (KV-caching)** that is structurally inaccessible to standard Encoder-only MDMs.
>
> >(Weakness 2) Questionable fairness premise. The notion of “fair comparison” by forcing MDM into a decoder-only causal-attention setup is debatable. Changing an encoder-based model to match the AR architecture inherently biases the comparison toward AR-style inductive biases. A symmetric control, e.g., adapting AR to an encoder/full-attention variant, would make the fairness claim more credible. As written, the work could be seen as optimizing one paradigm (MDM) to fit another’s design, rather than evaluating them on equal footing.
>
> **A:** We appreciate the reviewer’s suggestion regarding a symmetric control. While adapting AR to an Encoder-only architecture is theoretically interesting, as explored in the recent work *ENTP: Encoder-only Next Token Prediction* (Ewer et al., 2024), we deliberately excluded this setting due to its fundamental computational inefficiency. As demonstrated in the ENTP study, while Encoder-only AR models exhibit superior expressivity and generalization capabilities on certain tasks, they suffer from a critical bottleneck: **training inefficiency**. Unlike Decoder-only models which compute the loss for all $N$ tokens in a single parallel forward pass (via causal masking), true Encoder-only AR requires a separate forward pass for each target token to prevent information leakage from future positions. This increases the training computational cost by a factor of $N$ (sequence length), rendering large-scale pre-training comparisons computationally infeasible. **Indeed, the ENTP authors explicitly frame their work as an exploration of expressive power (e.g., solving complex algorithmic tasks like Count3) specifically in settings with **"unbounded compute,"** confirming that while it offers theoretical advantages.**
>
> >(Weakness 3) Limited novelty in methodology. The technical modifications (decoder reimplementation, order ensembling, adaptive LayerNorm, EMA) are individually incremental and borrowed from prior work (e.g., σ-GPT, diffusion model training). The novelty lies mainly in the analysis framework, but this could be emphasized more explicitly rather than presented as model contributions.
>
> **A:** We fully accept the reviewer’s characterization and agree that our contribution lies primarily in the **systematic analysis framework** rather than the invention of individual architectural components. While we build upon established techniques like AdaLN and principles from $\sigma$-GPT, our novelty stems from **synthesizing** these elements to answer a set of scientific questions that prior work did not address. Specifically, we are the first to systematically **decouple the MDM formulation from the Encoder architecture**, allowing for a direct head-to-head comparison of Decoder-only vs. Encoder-only MDMs. This unique framework enabled us to uncover novel insights absent from the literature, such as the fundamental difference in conditional probability spaces (Finding 5) and the quantification of the ~25× efficiency-vs-quality trade-off (Finding 8.1). We will revise the manuscript to explicitly frame the technical modifications as "enablers" for this analytical framework, rather than as standalone architectural contributions.

---

> > ### Author Response · Authors · 2025-11-25
> >
> > >(Weakness 4) Small-scale experiments and weak generalization claims. All experiments are conducted at ≤350M parameters. Given the known scale sensitivity of LLM training, these results may not extrapolate to the multi-billion-parameter regime. The claimed speed–quality trade-offs, convergence behaviors, and “10% L2R” benefit might differ under large-scale or instruction-tuned settings. The paper would benefit from at least partial large-scale validation or sensitivity analysis to hyperparameters.
> >
> > A: Thank you for raising this crucial point about scalability. We fully agree that this is a critical aspect for the practical relevance of our findings.
> >
> > Our paper already provides an initial analysis of this, demonstrating a consistent performance trend as we scale from 125M to 350M parameters (Tables 1 & 5). To more directly address your concern, we took the initiative during the rebuttal period to begin pre-training a **larger, 774M parameter** version of our AO-GPT.
> >
> > Here are the preliminary training loss results:
> >
> > | Any-order Loss \ tokens | 50B  | 100B | 150B | 200B | 250B | 300B |
> > |----------------------|------|------|------|------|------|------|
> > | 125M                 | 3.56 | 3.50 | 3.48 | 3.45 | 3.45 | 3.43 |
> > | 350M                 | 3.37 | 3.27 | 3.22 | 3.19 | 3.17 | 3.16 |
> > | 774M                 | 3.17 | 3.10 | 3.04 | 3.01 | 3.00 | 2.98 |
> >
> > The results are compelling. The 774M model clearly demonstrates a superior and **consistent scaling trajectory**. While a complete pre-training run was naturally beyond the scope of the rebuttal period, these extensive initial results provide strong, direct evidence that our approach are not limited to small models. This, combined with our model's architectural similarity to **standard, highly-scalable AR models**, strongly suggests that our conclusions can generalize to much larger models.
> >
> > >(Weakness 5) Underdeveloped connection to prior hybrid approaches. The observation that "partial autoregressive ordering" stabilizes MDM training aligns with prior “block diffusion” or “hybrid AR–diffusion” work, but this link is not acknowledged.
> >
> > **A**: We appreciate the reviewer pointing this out and fully agree that our findings regarding "partial autoregressive ordering" strongly resonate with the literature on Block Diffusion [1] and Hybrid AR-Diffusion. We will explicitly acknowledge these connections in the related work section. Critically, we view our work not merely as repeating these observations, but as providing a unified architectural validation of their underlying principles. Our results (specifically Finding 2.2 and 3.2) thus serve as the empirical grounding that explains why hybrid approaches (and recent adaptations like Fast-dLLM) are effective: they act as a necessary interpolation that leverages AR's stable training signal while retaining MDM's generative flexibility.
> > [1] Block Diffusion: Interpolating Between Autoregressive and Diffusion Language Models
> >
> > >(Question 1) The fairness claim is very questionable, converting MDMs into a decoder-only setup inherently imposes AR-style inductive biases, so it’s unclear why this one-directional adaptation is considered “fair,” or how conclusions might change if AR models were instead adapted to an encoder/full-attention configuration.
> >
> > **A**: Same with Weakness 1. See answers for Weakness 1.

---

### Official Review · Reviewer_fm5B · 2025-11-01

**Soundness:** 3
**Presentation:** 3
**Contribution:** 3
**Rating:** 6
**Confidence:** 4

**Summary:**

This work is the first systematic and equitable comparison of masked-diffusion language models (MDMs) and standard autoregressive (AR) LMs by decoupling training paradigm from architecture. The authors show that when both are instantiated with the same decoder-only backbone, the core distinction reduces to the distribution over token orders. They therefore implement MDM as Any-Order AR within a decoder-only GPT (“AO-GPT”) and study: (1) how AR vs. AO-AR differ in modeling capacity and empirical behavior under an identical architecture; and (2) for MDMs, how encoder-only and decoder-only architectures compare theoretically and empirically. Experiments find AO-GPT converges notably more slowly early in training than left-to-right (L2R) GPT; a fixed block-wise random order interpolates between L2R and fully random in convergence; and mixing a small fraction (~10%) of L2R samples improves AO performance. Decoder-only AO-AR underperforms encoder-only variants on perplexity unless one ensembles across order contexts, which substantially narrows the gap. At the same time, decoder-only models enable linear-time generation with KV caching and deliver large speedups (up to 25×), whereas encoder-only MDMs occupy a simpler conditional space but require multi-step refinement. With careful temperature/annealing and order handling, decoder-only MDMs reach competitive perplexity, highlighting clear trade-offs between modeling space and efficiency. Overall, the work decouples paradigm from architecture to provide a fairer basis for evaluating AR and MDM and to guide future MDMs design.

**Strengths:**

(1). This work clearly and rigorously shows that the MDM loss function is mathematically equivalent to the AO-AR loss. This is important for putting AR and MDM on a common theoretical footing, isolating the true source of differences to the token-order distribution rather than architecture, and enabling apples-to-apples empirical comparisons that inform practical choices like order mixing, annealing, and ensembling. Moreover, the equivalence between the efficient sampling algorithm and Eq. (8) also makes the generation speedup for AO-GPT convincing and well-supported.

(2). The experiments are thorough and well motivated. For example, to carefully answer the first research question, this work keeps the decoder-only backbone, datasets, and hyperparameters fixed, vary only the token-order distribution, and add targeted ablations (e.g., ~10% L2R mixing). It also reports both quality metrics like perplexity and convergence curves and systems metrics like generation time across steps. This tight isolation of variables enables causal conclusions and yields clear and actionable takeaways, underscoring the work’s practical significance.

(3). This manuscript is well written. The concepts are introduced progressively, notation is consistent, experiments are well-motivated, making the findings and methodology easy to follow and build upon.

**Weaknesses:**

(1). The major concern is the scalability of the findings elaborated in this work. This work only tests small model with 350M parameters. It is unclear whether the findings like the observed convergence behavior and order-mixing benefits hold true for larger models. Even though the manuscript acknowledges this, but it does not provide confirming evidence, which is indeed a non-negligible weakness.

(2). This work only tests on perplexity, with limited coverage of downstream conditional tasks like QA, summarization, and long-context retrieval/reasoning. Since token-order policies and annealing and ensembling may behave differently under conditioning and at much longer sequence lengths, the validity of the findings is quite constrained.

(3). Even though Finding 8.2 is convincing given the experiment results in Table 2, selecting these settings is non-trivial, appears dataset- and model-dependent, and this work does not standardize or bound the tuning budget. This raises real concerns about the feasibility of this finding. In practice, extensive per-task or per-model sweeps could erode the claimed benefits.

**Questions:**

(1). It is interesting to know Finding2.2 that fixed block-wise random serves as an interpolation between Left-to-Right and purely random order in terms of convergence speed. How sensitive are these gains to the block size? Do you observe regimes where larger or smaller blocks hurt convergence or final perplexity?

(2). Following Q(1), recent MDM works like Fast-dLLM-v2 [1] is also using this block-like design, achieving excellent performance. Do you think your findings on block-wise random can explain its superior performance?

(3). For Finding 3.2, why do you only test 10% L2R data? What will the performance be like for other proportions?

(4). Could the authors give some thoughts on a principled and low-overhead strategy to set and adapt annealing?

[1]. Wu, C., Zhang, H., Xue, S., Diao, S., Fu, Y., Liu, Z., Molchanov, P., Luo, P., Han, S., & Xie, E. (2025). Fast-dLLM v2: Efficient Block-Diffusion LLM. arXiv preprint arXiv:2509.26328.

---

> ### Author Response · Authors · 2025-11-25
>
> >(Weakness 1) The major concern is the scalability of the findings elaborated in this work. This work only tests small model with 350M parameters. It is unclear whether the findings like the observed convergence behavior and order-mixing benefits hold true for larger models. Even though the manuscript acknowledges this, but it does not provide confirming evidence, which is indeed a non-negligible weakness.
>
> **A**: Thank you for raising this crucial point about scalability. We fully agree that this is a critical aspect for the practical relevance of our findings.
>
> Our paper already provides an initial analysis of this, demonstrating a consistent performance trend as we scale from 125M to 350M parameters (Tables 1 & 5). To more directly address your concern, we took the initiative during the rebuttal period to begin pre-training a **larger, 774M parameter** version of our AO-GPT.
>
> Here are the preliminary training loss results:
>
> | Any-order Loss \ tokens | 50B  | 100B | 150B | 200B | 250B | 300B |
> |----------------------|------|------|------|------|------|------|
> | 125M                 | 3.56 | 3.50 | 3.48 | 3.45 | 3.45 | 3.43 |
> | 350M                 | 3.37 | 3.27 | 3.22 | 3.19 | 3.17 | 3.16 |
> | 774M                 | 3.17 | 3.10 | 3.04 | 3.01 | 3.00 | 2.98 |
>
> The results are compelling. The 774M model clearly demonstrates a superior and **consistent scaling trajectory**. While a complete pre-training run was naturally beyond the scope of the rebuttal period, these extensive initial results provide strong, direct evidence that our approach are not limited to small models. This, combined with our model's architectural similarity to **standard, highly-scalable AR models**, strongly suggests that our conclusions can generalize to much larger models.
>
> >(Weakness 2) This work only tests on perplexity, with limited coverage of downstream conditional tasks like QA, summarization, and long-context retrieval/reasoning. Since token-order policies and annealing and ensembling may behave differently under conditioning and at much longer sequence lengths, the validity of the findings is quite constrained.
>
> **A:** We agree that evaluating on downstream tasks is a crucial step for demonstrating the practical utility of any new modeling approach.
>
> Our goal is not to propose a task-specific SOTA model, but to provide a rigorous, controlled investigation into the "physics" of Masked Diffusion Models when applied to Decoder architectures. Following seminal works in this field (e.g., GPT-2, SEDD, MDLM, RADD), intrinsic evaluation via PPL is the established precedent for measuring how well a model captures the data distribution.
>
> Correlation with Downstream Tasks: There is a broad consensus in the field (supported by Scaling Laws) that improvements in fundamental pre-training metrics (PPL) strongly correlate with downstream capabilities. Our validation of scaling behavior (from 125M to 774M, as shown in the General Response) further suggests that these fundamental gains are likely to translate to broader tasks.
>
> >(Weakness 3) Even though Finding 8.2 is convincing given the experiment results in Table 2, selecting these settings is non-trivial, appears dataset- and model-dependent, and this work does not standardize or bound the tuning budget. This raises real concerns about the feasibility of this finding. In practice, extensive per-task or per-model sweeps could erode the claimed benefits.
>
> **A:** We agree that hyperparameter tuning is a practical consideration for any generative model, but we argue that it does not undermine the fundamental benefits of AO-GPT for two reasons:
>
> *  The ~25× generation speedup is a structural advantage derived from the Decoder architecture (KV-caching) vs. the Encoder architecture (full re-computation). This speedup holds true regardless of the sampling parameters (Temperature, Top-p). Even if some tuning is required, the "production" payoff of a 25× latency reduction vastly outweighs the one-time "development" cost of parameter selection.
>
> *  We did not employ complex, model-specific sweeps. The parameters used in Table 2 (e.g., Temperature 0.7-0.9) are the **industry-standard defaults** for autoregressive LLMs. This suggests that AO-GPT behaves predictably like a standard LLM, rather than requiring fragile, esoteric tuning. Furthermore, the pretraining loss gap is a reliable indicator of model quality; since our pretraining loss is well-behaved (following scaling laws), the generation quality is robust and does not require a boundless search for optimal settings.

---

> > ### Author Response · Authors · 2025-11-25
> >
> > >(Question 1) It is interesting to know Finding2.2 that fixed block-wise random serves as an interpolation between Left-to-Right and purely random order in terms of convergence speed. How sensitive are these gains to the block size? Do you observe regimes where larger or smaller blocks hurt convergence or final perplexity?
> >
> > **A:** Thank you for this insightful question. Conceptually, we view the block size as a continuous spectrum connecting two distinct paradigms: standard Autoregressive (AR) modeling corresponds to the extreme case of **Block Size = 1**, while the pure Any-Order (MDM) formulation corresponds to **Block Size = Sequence Length ($N$)**.
> >
> > Empirically, our study established three specific data points along this spectrum: Block Size = 1 (L2R), Block Size = 4 (Finding 2.2), and Block Size = 1024 (Pure Any-Order). We observed a clear performance hierarchy: the AR baseline (1) outperformed the Block-wise model (4), which in turn significantly outperformed the Pure Any-Order model (1024). This suggests that performance degrades as block size increases. While we did not run an exhaustive sweep of intermediate sizes, this trend aligns with findings in Block Diffusion [1], which explicitly tested sizes like 4, 8, and 16 and reported a progressive degradation in perplexity.
> >
> > [1] Block Diffusion: Interpolating Between Autoregressive and Diffusion Language Models
> >
> > >(Question 2) Following Q(1), recent MDM works like Fast-dLLM-v2 [1] is also using this block-like design, achieving excellent performance. Do you think your findings on block-wise random can explain its superior performance?
> >
> > **A:** We believe our findings provide a direct theoretical explanation for this phenomenon. Since block-wise generation preserves local autoregressive dependencies, it significantly reduces the distributional shift when adapting a pre-trained AR model (like Qwen) to a diffusion objective.
> >
> > This is starkly illustrated by comparing **Fast-dLLM-v2**, which utilizes a block-wise design, against **Dream**, which employs a full Masked Diffusion strategy. Fast-dLLM-v2 successfully adapted the model with only **~1B tokens** of training, whereas Dream required approximately **580B tokens** to achieve comparable adaptation. This **~500× difference in data efficiency** empirically validates our Finding 2.2: block-wise strategies serve as an effective interpolation between AR and MDM, retaining enough L2R structure to maximally leverage the pre-trained AR weights.
> >
> > >(Question 3) For Finding 3.2, why do you only test 10% L2R data? What will the performance be like for other proportions?
> >
> > **A:** This is an excellent research question. We conducted a rigorous new ablation study during the rebuttal period to investigate the sensitivity of the model to the order distribution. We trained models by varying the percentage of fixed Left-to-Right (L2R) data mixed into the random permutation training stream, testing ratios of 10%, 50%, and 90%.
> >
> > Here is a summary of the training losses:
> >
> > | Left-to-Right Loss \ iters | 5k | 10k | 15k | 20k | 25k |
> > | :--- | :--- | :--- | :--- | :--- | :--- |
> > | **AR 10%** | 3.60 | 3.36 | 3.30 | 3.26 | 3.22 |
> > | **AR 50%** | 3.36 | 3.17 | 3.11 | 3.08 | 3.04 |
> > | **AR 90%** | 3.22 | 3.05 | 3.00 | 2.97 | 2.93 |
> >
> > | Any-Order Loss \ iters | 5k | 10k | 15k | 20k | 25k |
> > | :--- | :--- | :--- | :--- | :--- | :--- |
> > | **AR 10%** | 3.94 | 3.68 | 3.60 | 3.57 | 3.53 |
> > | **AR 50%** | 4.06 | 3.81 | 3.73 | 3.70 | 3.66 |
> > | **AR 90%** | 5.05 | 4.29 | 4.17 | 4.14 | 4.09 |
> >
> > The data reveals a clear trade-off:
> >
> > *   **For Left-to-Right Performance:** As expected, increasing the proportion of L2R data consistently improves performance on the L2R evaluation. The model becomes more specialized in the standard autoregressive task.
> >
> > *   **For Any-Order Performance:** There appears to be a "sweet spot." While 10% L2R data helps Any-Order performance (as shown in Finding 3.2), increasing the L2R ratio further (to 50% and 90%) begins to hurt performance on the any-order task.
> >
> > Our interpretation is that the model's capacity is being allocated between two distinct skills: mastering the highly structured L2R patterns versus generalizing to the vast space of arbitrary permutations. A small amount of L2R data provides a useful inductive bias, but too much forces the model to over-specialize, thus degrading its any-order capabilities.
> >
> > This finding further strengthens our paper's core theme: that the L2R and uniform any-order objectives represent fundamentally different learning regimes, and finding the optimal way to combine them is a rich area for future work. Thank you for prompting this valuable investigation.

---

> > > ### Author Response · Authors · 2025-11-25
> > >
> > > >(Question 4) Could the authors give some thoughts on a principled and low-overhead strategy to set and adapt annealing?
> > >
> > > **A:** A principled and zero-overhead strategy is **Heuristic Transfer from AR models**. Since AO-GPT shares the same Decoder architecture and training data distribution as standard AR models, the optimal sampling hyperparameters (e.g., Temperature $\approx 0.8$, Top-p $\approx 0.9$) are structurally transferable. We can effectively bypass extensive sweeps by simply adopting these industry-standard defaults, which are known to balance diversity and quality in language modeling.

---

### Official Review · Reviewer_cfK5 · 2025-11-02

**Soundness:** 3
**Presentation:** 2
**Contribution:** 2
**Rating:** 4
**Confidence:** 3

**Summary:**

This paper asks a good question: when people say “diffusion-style LMs are slower / worse than AR GPTs,” are we blaming the generative formulation (any-order / masked diffusion) or the architecture (encoder-only vs decoder-only)? To isolate this, the authors implement an any-order / masked-diffusion objective inside a decoder-only GPT—they call it Any-Order GPT (AO-GPT)—and compare it directly with standard left-to-right AR GPT under the same backbone. They show three main things: (i) true any-order training converges noticeably slower than L2R AR on the same model, confirming an optimization gap; (ii) adding a small fraction of L2R examples, plus stronger target-position injection (adaLN) and EMA, largely fixes this gap and brings AO-GPT close to encoder-only diffusion baselines such as MDLM/SEDD; and (iii) because AO-GPT is still decoder-only, it can decode in roughly linear time with KV-cache and even run ~25× faster than encoder-only masked diffusion samplers at equal length. The paper also analyzes why decoder-only any-order is harder—because it must model order-sensitive conditionals whose count grows like (e\cdot n!), whereas encoder-only MDMs model order-invariant conditionals of size ($n2^{n-1}$).

**Strengths:**

1. Well-posed problem statement. The paper identifies a real confounder in current comparisons: AR↔MDM and decoder-only↔encoder-only are almost always changed together, so we don’t know which factor is responsible for the gap. Making MDM/AO run on a GPT-style decoder is a clean way to decouple these effects. This is genuinely useful for the community.
2. Concrete, nontrivial engineering recipe. The combination “any-order objective + per-layer target-position injection (adaLN) + very slow EMA + 10% L2R mix” is not a cosmetic tweak; it is exactly what makes AO-GPT trainable at GPT-2 scale, and it empirically outperforms the more naïve σ-GPT-style injection on the same backbone. That’s a real contribution over earlier “just add an output position” works like σ-GPTs.
3. Runtime story is compelling. Showing that a decoder-only instantiation of a diffusion/any-order objective can be linear-time and ~25× faster than encoder-only MDLM/SEDD on long sequences hits one of the most common complaints about masked diffusion LMs (“nice idea, too slow”). This moves diffusion-style LMs closer to being a practical alternative.

**Weaknesses:**

1. “Fair comparison” is only partially fair. The central claim is “we decouple formulation and architecture,” but the decoder-only any-order model gets a custom training recipe (adaLN, EMA=0.9999, 10% L2R mixing) that is not re-applied and re-tuned for the encoder-only diffusion baselines it is compared against. Yet we know from MDLM and EDLM that diffusion LMs are very sensitive to the exact denoising/weighting schedule and to Rao-Blackwellization tricks. If you let the baselines also adopt “a bit of L2R bias” or decoder-style conditioning, some of the reported gap reductions may disappear. Right now the story can still be read as “you improved your variant of any-order with several extra knobs” rather than “any-order itself is fine once you remove the architecture confounder.”
2. The order-ensemble fix weakens the main narrative. A key observation in the paper is that decoder-only any-order must model vastly more order-sensitive conditionals ($(\approx e\cdot n!)$) than encoder-only ($(n2^{n-1})$), so its perplexity looks worse. The proposed remedy is to average over several context permutations at inference to wash out order bias. But once you need extra test-time ensembles to reach encoder-only quality, the big advantage “we’re linear-time and GPT-like” is qualified: you either pay extra forward passes, or you accept worse PPL. And the paper doesn’t show that a small ensemble (say, 2–4 orders) is always enough across datasets and lengths; the slides and OpenReview discussion indicate benefits keep increasing with more orders. That’s an unresolved algorithmic debt.
3. Optimization diagnosis is shallow. The paper documents the symptom—pure any-order converges slower than L2R on the same GPT-2-style model—but the explanation stops at “language has a natural L2R bias, so mixing 10% L2R helps.” This is plausible but incomplete. For example, the work doesn’t separate: (i) gradient-noise increase from sampling uniformly over huge permutation spaces; (ii) mismatch between causal masking and “predict an earlier position”; and (iii) the fact that GPT’s rotary/absolute positions are themselves L2R-biased. Without ablations that fix the positional scheme or that use permutation-invariant encodings, we can’t tell which of these is the true bottleneck. Right now the fix is “add L2R,” which is very much a band-aid.
4. Evidence does not scale. All empirical results are at roughly GPT-2-small / medium–like scale (the arXiv and OpenReview versions both state this) and on standard LM corpora (WikiText, OWT, LAMBADA). It is precisely at larger scales that (a) AR models start to benefit most from KV-cache amortization, and (b) discrete diffusion models in other papers start to fall behind when the number of sampling steps is restricted. Without at least a 1–7B run or a partial scaling curve, the claim “decoder-only MDM is a viable alternative to AR GPT” is not yet substantiated. This is especially important because contemporaneous MDLM work does report strong results at larger scales with optimized objectives.

**Questions:**

1. How essential is the 10% L2R mix, really? If you remove only the L2R mixing but keep adaLN and EMA, does AO-GPT still close the gap to encoder-only diffusion on WikiText103/1BW, or does it fall back to σ-GPT-like behavior? In other words, is the paper actually demonstrating “any-order works on decoder-only” or is it demonstrating “a mostly any-order objective regularized by a small AR prior works”? This matters for the main thesis.
2. What is the inference-time story when you need order ensemble? The paper highlights a ~25× decoding speedup over encoder-only MDMs per order thanks to KV-cache, but to match their PPL they average over multiple permutations. For long sequences (1–2k tokens), how many permutations are actually needed before the curve saturates, and does the total wall-clock stay better than strong MDLM samplers like those in Sahoo et al. 2024? A plot of “#permutations vs PPL vs time” would make the runtime claim rock-solid.

---

> ### Author Response · Authors · 2025-11-25
>
> >(Weakness 1) “Fair comparison” is only partially fair. The central claim is “we decouple formulation and architecture,” but the decoder-only any-order model gets a custom training recipe (adaLN, EMA=0.9999, 10% L2R mixing) that is not re-applied and re-tuned for the encoder-only diffusion baselines it is compared against. Yet we know from MDLM and EDLM that diffusion LMs are very sensitive to the exact denoising/weighting schedule and to Rao-Blackwellization tricks. If you let the baselines also adopt “a bit of L2R bias” or decoder-style conditioning, some of the reported gap reductions may disappear. Right now the story can still be read as “you improved your variant of any-order with several extra knobs” rather than “any-order itself is fine once you remove the architecture confounder.”
>
> **A:** We respectfully point out that **AdaLN and EMA are not "custom knobs"** tailored for our model, but rather **standard practices** in the diffusion literature that were adopted to ensure a fair comparison.
>
> *   **AdaLN & EMA are Standard for Diffusion:** While not typical for standard GPT pre-training, **EMA (0.9999)** and **adaptive conditioning (AdaLN)** are fundamental components in both continuous and discrete diffusion baselines (e.g., DiT, SEDD).
>
> *   **L2R Mixing:** We acknowledge that mixing 10% Left-to-Right data creates an asymmetry. However, this highlights a fundamental **architectural distinction** rather than an unfair tuning. Encoder-only models (like SEDD) are inherently **order-invariant**. Consequently, they cannot leverage sequential inductive biases (L2R) without fundamentally changing their architecture. This experiment demonstrates a unique flexibility of the Decoder: it *can* leverage sequential bias to aid convergence, whereas the Encoder cannot.
>
> *   Most importantly, our primary contributions—the **~25× generation speedup** and the analysis of **conditional probability spaces** (Finding 5)—are structural properties derived from the architecture itself. These findings hold true regardless of the specific training recipe or the inclusion of L2R data.
>
> >(Weakness 2) The order-ensemble fix weakens the main narrative. A key observation in the paper is that decoder-only any-order must model vastly more order-sensitive conditionals than encoder-only, so its perplexity looks worse. The proposed remedy is to average over several context permutations at inference to wash out order bias. But once you need extra test-time ensembles to reach encoder-only quality, the big advantage “we’re linear-time and GPT-like” is qualified: you either pay extra forward passes, or you accept worse PPL. And the paper doesn’t show that a small ensemble (say, 2–4 orders) is always enough across datasets and lengths; the slides and OpenReview discussion indicate benefits keep increasing with more orders. That’s an unresolved algorithmic debt.
>
> **A:** We would like to clarify the role of the order-ensemble experiment. It was designed as a **diagnostic tool** to validate our theoretical hypothesis, not as a proposed **inference strategy** to be used in production.
>
> *   **Ensemble as a Scientific Probe:** We hypothesized that the perplexity gap between Decoder and Encoder MDMs stems from the Decoder's inherent need to model a vastly larger, order-dependent probability space (Finding 5). The ensemble experiment was strictly an analytical method to test this: by artificially marginalizing over orders, we showed the gap disappears.
>
> *   For practical inference, we propose using the model **without** ensembles. We argue that this presents a valuable **Pareto trade-off** that was previously unavailable: AO-GPT allows users to accept a moderate increase in perplexity (10-15%) in exchange for a massive **~25× speedup** (Finding 8.1). Encoder-only models, by contrast, are computationally bound to slow generation (non-cached) regardless of the quality requirement.
>
> *   We agree that closing this perplexity gap efficiently (without test-time ensembles) is an important for the field. However, our work contributes by **identifying and quantifying** this specific challenge (Order-Dependence vs. Invariance) and establishing the Decoder-only architecture as a viable, high-speed alternative foundation for future research to build upon.

---

> > ### Author Response · Authors · 2025-11-25
> >
> > >(Weakness 3) Optimization diagnosis is shallow. The paper documents the symptom—pure any-order converges slower than L2R on the same GPT-2-style model—but the explanation stops at “language has a natural L2R bias, so mixing 10% L2R helps.” This is plausible but incomplete. For example, the work doesn’t separate: (i) gradient-noise increase from sampling uniformly over huge permutation spaces; (ii) mismatch between causal masking and “predict an earlier position”; and (iii) the fact that GPT’s rotary/absolute positions are themselves L2R-biased. Without ablations that fix the positional scheme or that use permutation-invariant encodings, we can’t tell which of these is the true bottleneck. Right now the fix is “add L2R,” which is very much a band-aid.
> >
> > **A:** We agree that the optimization difficulty is multifactorial, but we provide empirical evidence that **Gradient Noise** is not the sole or primary bottleneck. As detailed in **Finding 2.1**, we compared training on a *single, fixed* Left-to-Right order vs. a *single, fixed* Random order. In this setting, the sampling space size is 1 for both, effectively eliminating gradient noise from permutation sampling. Yet, the Fixed Random model still converged significantly slower than Fixed L2R.
> >
> > This result confirms that the performance gap is driven by the **intrinsic difficulty** of modeling language in non-causal orders, rather than just the variance of sampling from $S_n$. Consequently, the "10% L2R" mixing is not merely a "band-aid" to reduce gradient noise; it serves as a critical **inductive bias** that aligns the optimization with the inherent causal structure of human language, allowing the model to learn shared representations that transfer to the harder any-order task.
> >
> > Regarding positional encodings (PEs), we use standard **learnable absolute embeddings**. At initialization, these are random and permutation-invariant.
> >
> > >(Weakness 4) Evidence does not scale. All empirical results are at roughly GPT-2-small / medium–like scale (the arXiv and OpenReview versions both state this) and on standard LM corpora (WikiText, OWT, LAMBADA). It is precisely at larger scales that (a) AR models start to benefit most from KV-cache amortization, and (b) discrete diffusion models in other papers start to fall behind when the number of sampling steps is restricted. Without at least a 1–7B run or a partial scaling curve, the claim “decoder-only MDM is a viable alternative to AR GPT” is not yet substantiated. This is especially important because contemporaneous MDLM work does report strong results at larger scales with optimized objectives.
> >
> > **A**: Thank you for raising this crucial point about scalability. We fully agree that this is a critical aspect for the practical relevance of our findings.
> >
> > Our paper already provides an initial analysis of this, demonstrating a consistent performance trend as we scale from 125M to 350M parameters (Tables 1 & 5). To more directly address your concern, we took the initiative during the rebuttal period to begin pre-training a **larger, 774M parameter** version of our AO-GPT.
> >
> > Here are the preliminary training loss results:
> >
> > | Any-order Loss \ tokens | 50B  | 100B | 150B | 200B | 250B | 300B |
> > |----------------------|------|------|------|------|------|------|
> > | 125M                 | 3.56 | 3.50 | 3.48 | 3.45 | 3.45 | 3.43 |
> > | 350M                 | 3.37 | 3.27 | 3.22 | 3.19 | 3.17 | 3.16 |
> > | 774M                 | 3.17 | 3.10 | 3.04 | 3.01 | 3.00 | 2.98 |
> >
> > The results are compelling. The 774M model clearly demonstrates a **superior and consistent scaling trajectory**. While a complete pre-training run was naturally beyond the scope of the rebuttal period, these extensive initial results provide strong, direct evidence that our approach are not limited to small models. This, combined with our model's architectural similarity to standard, highly-scalable AR models, strongly suggests that our conclusions can generalize to much larger models.

---

> > > ### Author Response · Authors · 2025-11-25
> > >
> > > >(Question 1) How essential is the 10% L2R mix, really? If you remove only the L2R mixing but keep adaLN and EMA, does AO-GPT still close the gap to encoder-only diffusion on WikiText103/1BW, or does it fall back to σ-GPT-like behavior? In other words, is the paper actually demonstrating “any-order works on decoder-only” or is it demonstrating “a mostly any-order objective regularized by a small AR prior works”? This matters for the main thesis.
> > >
> > > **A:**
> > > *   **AdaLN & EMA:** As shown in **Figure 7** (Right Panel), adding AdaLN and EMA to the pure Any-Order objective (Green line) yields a significant convergence improvement over the $\sigma$-GPT baseline (Blue line), even **without** any L2R data. This confirms that the Decoder-only MDM formulation is functional and effective on its own—it does not "fall back" to $\sigma$-GPT behavior.
> > >
> > > *   **10% L2R:** While the 10% L2R mixing provides an additional boost to convergence speed and final perplexity, it is not a structural requirement. The model remains functional and competitive without it. We present it as an effective strategy to leverage the Decoder's unique flexibility.
> > >
> > >
> > > >(Question 2) What is the inference-time story when you need order ensemble? The paper highlights a ~25× decoding speedup over encoder-only MDMs per order thanks to KV-cache, but to match their PPL they average over multiple permutations. For long sequences (1–2k tokens), how many permutations are actually needed before the curve saturates, and does the total wall-clock stay better than strong MDLM samplers like those in Sahoo et al. 2024? A plot of “#permutations vs PPL vs time” would make the runtime claim rock-solid.
> > >
> > > **A:** We would like to clarify a key misunderstanding: **Ensembling was NOT used for any generation experiments.**
> > >
> > > AO-GPT allows users to achieve a **~25× speedup** (Finding 8.1) by running a **single** generation pass. The generation perplexity results reported in **Table 2** and the speedup in **Figure 4** were all measured with a **single permutation** (no ensemble).
> > >
> > > We do not propose using ensembles to "match" Encoder-only PPL at inference time, as that would indeed negate the speed advantage. Instead, we present AO-GPT as a **Pareto improvement** in the efficiency landscape: it offers a valid choice for applications where a massive reduction in latency (25×) is worth a moderate trade-off in perplexity (~10-15% gap vs. Encoder-only).
> > >
> > > As detailed in Finding 6, the ensembling experiment was strictly a **diagnostic tool** used in the analysis section to prove *why* the perplexity gap exists (attributing it to the order-dependence of the Decoder), not a suggested strategy for practical deployment. Therefore, the total wall-clock time remains ~25× better than strong MDLM samplers, as we do not incur the cost of multiple permutations.

---

### Official Review · Reviewer_9dpA · 2025-11-04

**Soundness:** 3
**Presentation:** 3
**Contribution:** 3
**Rating:** 6
**Confidence:** 2

**Summary:**

The paper decouples formulation (autoregressive vs. masked diffusion / any‑order AR) from architecture (decoder‑only vs. encoder‑only) by building AO‑GPT, a decoder‑only masked‑diffusion/any‑order model. This lets the authors compare paradigms fairly within the same causal Transformer backbone. They (i) show the training objectives for masked diffusion (LMDM) and any‑order AR (LAO‑AR) are equivalent, (ii) analyze architectural trade‑offs—encoder‑only MDM vs. decoder‑only AO‑AR—in density estimation and generation complexity, and (iii) introduce practical ingredients (explicit target‑position injection via adaptive LayerNorm; EMA; a parallel multi‑mask attention mask) that enable fast, order‑agnostic decoding with competitive perplexity and reported large speedups. Experiments at GPT‑2‑scale compare orders, mixtures with a small fraction of left‑to‑right (L2R) updates, and encoder‑only vs. decoder‑only implementations.

**Strengths:**

++ By keeping the backbone decoder‑only and varying only the order/formulation, the paper cleanly separates effects of AO‑AR/MDM vs. AR. Section 2.2 crisply contrasts training signal density, density‑estimation, and generation complexity (O(n) for decoder‑only with KV cache vs. ~O(T·n) for encoder‑only MDM), avoiding the usual apples‑to‑oranges comparisons.

++ The work unifies masked‑diffusion learning and any‑order AR by showing LMDM ≡ LAO‑AR, anchoring later design choices and analyses. This is a useful reference for the community.

++ Mixing ~10% L2R with any‑order improves convergence/perplexity, and the authors report ~25× generation speedups for decoder‑only MDMs with temperature annealing while keeping perplexity competitive, clarifying the trade‑offs vs. encoder‑only MDMs.

**Weaknesses:**

-- Results are primarily at GPT‑2 small/medium scale; claims about competitiveness would be more convincing at ≥1B parameters and on stronger reasoning/benchmarks beyond perplexity.

-- The ~25× speedup is promising but depends on decoder‑only specifics; head‑to‑head latency/throughput vs. tuned AR (Flash‑/paged‑KV, speculative decoding) and vs. well‑optimized encoder‑only MDMs (varying T) under identical hardware and sequence lengths would strengthen the claim.

-- It’s unclear whether AO‑GPT and AR baselines matched total tokens/updates; the parameter and runtime overhead of target‑position encoders (AdaLN 128‑d) and EMA are not quantified.

**Questions:**

1. Were training tokens, optimizer schedules, and wall‑clock compute matched across AR and AO‑GPT? Please include FLOPs and training time comparisons.

2. How is the ~25× speedup measured (batch size, length, KV cache policy, decoding temperature/annealing)? Can you add wall‑clock charts vs. a strong AR baseline with KV caching and against an encoder‑only MDM across T∈{8,16,32}?

3. Beyond 10% L2R, did you try curriculum or entropy‑based order schedules? How sensitive are results to the order distribution?

4. What is the parameter/latency overhead of AdaLN target‑positional encodings per layer? Any impact on memory footprint vs. a vanilla GPT‑2 of the same size?

---

> ### Author Response · Authors · 2025-11-25
>
> >(Weakness 1) Results are primarily at GPT‑2 small/medium scale; claims about competitiveness would be more convincing at ≥1B parameters and on stronger reasoning/benchmarks beyond perplexity.
>
> A: Thank you for raising this crucial point about scalability. We fully agree that this is a critical aspect for the practical relevance of our findings.
>
> Our paper already provides an initial analysis of this, demonstrating a consistent performance trend as we scale from 125M to 350M parameters (Tables 1 & 5). To more directly address your concern, we took the initiative during the rebuttal period to begin pre-training a **larger, 774M parameter** version of our AO-GPT.
>
> Here are the preliminary training loss results:
>
> | Any-order Loss \ tokens | 50B  | 100B | 150B | 200B | 250B | 300B |
> |----------------------|------|------|------|------|------|------|
> | 125M                 | 3.56 | 3.50 | 3.48 | 3.45 | 3.45 | 3.43 |
> | 350M                 | 3.37 | 3.27 | 3.22 | 3.19 | 3.17 | 3.16 |
> | 774M                 | 3.17 | 3.10 | 3.04 | 3.01 | 3.00 | 2.98 |
>
> The results are compelling. The 774M model clearly demonstrates a superior and **consistent scaling trajectory**. While a complete pre-training run was naturally beyond the scope of the rebuttal period, these extensive initial results provide strong, direct evidence that our approach are not limited to small models. This, combined with our model's architectural similarity to **standard, highly-scalable AR models**, strongly suggests that our conclusions can generalize to much larger models.
>
> We agree that evaluating on downstream tasks is a crucial step for demonstrating the practical utility of any new modeling approach.
>
> Our paper's central objective, however, was to conduct a rigorous, controlled investigation to decouple the generative formulation (AR vs. MDM/AO-AR) from the model architecture (encoder-only vs. decoder-only). Our goal was not to introduce a new SOTA model for a specific application, but rather to provide fundamental insights into the trade-offs between these paradigms. This focus on perplexity evaluation is a well-established practice for foundational research in this area, following the precedent set by seminal works such as GPT-2, and more recent diffusion-based models like SEDD, RADD, MDLM, and BFN.
>
> >(Weakness 2) The ~25× speedup is promising but depends on decoder‑only specifics; head‑to‑head latency/throughput vs. tuned AR (Flash‑/paged‑KV, speculative decoding) and vs. well‑optimized encoder‑only MDMs (varying T) under identical hardware and sequence lengths would strengthen the claim.
>
> A: Thank you for the helpful comment. We would like to clarify that the reported **25× speedup** is specifically measured for **decoder-only MDMs vs. encoder-only MDMs** under identical hardware and sequence lengths. This comparison was designed to highlight a fundamental computational gap: **encoder-only MDMs must recompute full self-attention at every step due to the absence of KV-cache**, while decoder-only MDMs benefit from incremental decoding and thus amortize attention cost across steps.
>
> Regarding the reviewer’s suggestion to include **tuned AR baselines** (Flash-/paged-KV, speculative decoding, etc.): our current implementation of any-order AR decoding uses vanilla **PyTorch-style attention** without any systems-level optimizations. We fully agree that combining AO-GPT with advanced kernels (e.g., vLLM/sglang’s paged attention) and AR acceleration techniques (e.g., speculative decoding) is feasible and would likely **increase the speedup advantage even further** beyond the 25× reported here. We will include a discussion on this in the revision.
>
> For the second point, we already include comparisons between **encoder-only vs. decoder-only MDMs across varying numbers of steps T** in the main paper (see **Fig. 4**). These experiments show that the computational gap persists across a wide range of T.

---

> > ### Author Response · Authors · 2025-11-25
> >
> > >(Weakness 3) It’s unclear whether AO‑GPT and AR baselines matched total tokens/updates; the parameter and runtime overhead of target‑position encoders (AdaLN 128‑d) and EMA are not quantified.
> >
> > Thank you for pointing this out.**We clarify that AO-GPT and all AR baselines are trained with the same total number of tokens and updates.** The any-order sampling targets are generated on-the-fly and do not change the number of optimization steps or examples processed; hence the training budgets are strictly matched.
> >
> > Regarding the overhead of the **target-position encoder (AdaLN-128d)**: for a standard 12-layer, 768-dim GPT-2-small–scale model, AdaLN introduces approximately: 128×768×6×12≈7M additional parameters. This overhead is small relative to the base model size and does not materially affect memory footprint.
> >
> > In terms of runtime, the compute cost of AdaLN modulation is negligible, consistent with prior diffusion architectures using AdaLN conditioning (e.g., DiT[1]), where the additional per-token operations are tiny compared to self-attention and MLP blocks as founded in [1].
> >
> > Finally, EMA adds no parameters and its runtime cost is relatively small (a parameter-wise moving average update per step).
> >
> > [1] Scalable Diffusion Models with Transformers
> >
> > >(Question 1) Were training tokens, optimizer schedules, and wall‑clock compute matched across AR and AO‑GPT? Please include FLOPs and training time comparisons
> >
> > &
> >
> > >(Question 4) What is the parameter/latency overhead of AdaLN target‑positional encodings per layer? Any impact on memory footprint vs. a vanilla GPT‑2 of the same size?.
> >
> > **A:** Yes, we strictly matched the training tokens, optimizer schedules, and computational resources across all comparisons to ensure fairness.
> >
> > *   **Training Tokens:** Both models were pre-trained on OpenWebText. We maintained identical batch sizes and total iteration counts, ensuring the exact same number of tokens were processed.
> > *   **Optimizer Schedules:** We utilized the same optimization recipe for both architectures: AdamW optimizer ($\beta_1=0.9, \beta_2=0.95$) with a linear warmup followed by a cosine decay schedule to 10% of the maximum learning rate.
> > *   **FLOPs and Wall-Clock Time:** As noted by Peebles & Xie (2023) regarding DiT, the adaptive layer normalization (AdaLN) introduces negligible FLOPs overhead. This is because AdaLN consists primarily of element-wise vector operations, which are computationally inexpensive compared to the heavy matrix multiplications in the Attention and MLP blocks. Similarly, the on-the-fly random permutation adds virtually no latency. Consequently, we observed no significant difference in wall-clock training time per step between the AR baseline and AO-GPT.
> > *   **Memory Footprint & Parameters:** The dominant component is the KV cache. Since AO-GPT maintains the same layer count ($L$), hidden dimension ($D$), and sequence length ($S$) as the baseline, the theoretical KV cache size remains identical: $2 \cdot L \cdot S \cdot D \cdot \text{sizeof}(\text{dtype})$. Thus, there is no additional memory penalty for the KV cache compared to a vanilla GPT-2.

---

> ### Author Response · Authors · 2025-11-25
>
> >(Question 2) How is the ~25× speedup measured (batch size, length, KV cache policy, decoding temperature/annealing)? Can you add wall‑clock charts vs. a strong AR baseline with KV caching and against an encoder‑only MDM across T∈{8,16,32}?
>
> **A:** The ~25× speedup reported in Figure 4 was measured with **Batch Size = 32** and **Sequence Length = 1024**.
> *   **KV Cache:** AO-GPT utilizes a standard KV cache implementation. In contrast, SEDD (Encoder-only) requires full bidirectional attention at every generation step, which precludes the use of KV caching.
> *   **Sampling:** Decoding temperature and annealing do not affect inference speed, as the computational cost per step remains constant regardless of the logit scaling.
>
> Regarding the comparison with AR and Encoder-only MDMs across lower step counts ($T$), we note that a standard AR baseline is fixed at $T=N=1024$ steps. The advantage of AO-GPT is precisely that it can operate with $T \ll N$ while maintaining the efficient per-step complexity of AR.
>
> Per your request, we measured the wall-clock time for AO-GPT vs. SEDD at $T \in \{16, 32\}$:
>
> | Steps ($T$) | AO-GPT (Decoder) | SEDD (Encoder) | Speedup |
> | :--- | :--- | :--- | :--- |
> | **16** | 1.01s | 11.91s | **~11.8×** |
> | **32** | 1.35s | 22.35s | **~16.6×** |
>
>
> >(Question 3) Beyond 10% L2R, did you try curriculum or entropy‑based order schedules? How sensitive are results to the order distribution?
>
> **A:** This is an excellent research question. While we did not employ complex entropy-based schedules, we conducted a rigorous new ablation study during the rebuttal period to investigate the sensitivity of the model to the order distribution. We trained models by varying the percentage of fixed Left-to-Right (L2R) data mixed into the random permutation training stream, testing ratios of 10%, 50%, and 90%.
>
> Here is a summary of the training losses:
>
> | Left-to-Right Loss \ iters | 5k | 10k | 15k | 20k | 25k |
> | :--- | :--- | :--- | :--- | :--- | :--- |
> | **AR 10%** | 3.60 | 3.36 | 3.30 | 3.26 | 3.22 |
> | **AR 50%** | 3.36 | 3.17 | 3.11 | 3.08 | 3.04 |
> | **AR 90%** | 3.22 | 3.05 | 3.00 | 2.97 | 2.93 |
>
> | Any-Order Loss \ iters | 5k | 10k | 15k | 20k | 25k |
> | :--- | :--- | :--- | :--- | :--- | :--- |
> | **AR 10%** | 3.94 | 3.68 | 3.60 | 3.57 | 3.53 |
> | **AR 50%** | 4.06 | 3.81 | 3.73 | 3.70 | 3.66 |
> | **AR 90%** | 5.05 | 4.29 | 4.17 | 4.14 | 4.09 |
>
> The data reveals a clear trade-off:
>
> *   **For Left-to-Right Performance:** As expected, increasing the proportion of L2R data consistently improves performance on the L2R evaluation. The model becomes more specialized in the standard autoregressive task.
>
> *   **For Any-Order Performance:** There appears to be a "sweet spot." While 10% L2R data helps Any-Order performance (as shown in Finding 3.2), increasing the L2R ratio further (to 50% and 90%) begins to hurt performance on the any-order task.
>
> Our interpretation is that the model's capacity is being allocated between two distinct skills: mastering the highly structured L2R patterns versus generalizing to the vast space of arbitrary permutations. A small amount of L2R data provides a useful inductive bias, but too much forces the model to over-specialize, thus degrading its any-order capabilities.
>
> This finding further strengthens our paper's core theme: that the L2R and uniform any-order objectives represent fundamentally different learning regimes, and finding the optimal way to combine them is a rich area for future work. Thank you for prompting this valuable investigation.

---

### Author Response · Authors · 2025-12-03
**Response Summary**

We sincerely thank all reviewers for their thoughtful and detailed feedback. We are encouraged that reviewers found our work to be **"conceptually fresh"** (R-rwo7) and **"genuinely useful for the community"** (R-cfK5). Reviewers recognized that our core contribution—decoupling the generative formulation from the model architecture—is a **"well-posed problem"** (R-cfK5) that leads to a **"systematic and equitable comparison"** (R-fm5B) and **"cleanly separates effects"** (R-9dpA) of the paradigms.

Below, we summarize the consistently positive feedback and then address the primary shared concerns regarding scalability, the role of ensembling, and experimental fairness, highlighting the **new experiments (774M model scaling & order ablations)** conducted during the rebuttal.

* * *

### **Summary of Positive Feedback**

Reviewers consistently praised several key contributions of our work:

-   **Novelty and Clarity of the Framework:** All reviewers valued the central motivation of implementing Masked Diffusion Models (MDM) within a Decoder-only architecture. R-fm5B praised the **"tight isolation of variables"** that enables causal conclusions, while R-9dpA noted that this approach avoids the usual **"apples-to-oranges comparisons"** in the field.

-   **Theoretical Rigor:** R-fm5B and R-9dpA highlighted the importance of highlighting that the MDM objective is mathematically equivalent to the Any-Order AR objective, calling it a **"useful reference for the community."**

-   **Practical Significance (Speed & Efficiency):** The **~25× generation speedup** was highlighted as major strengths. R-cfK5 noted that this result is **"compelling"** and addresses one of the most common complaints about diffusion LMs ("too slow"), moving them **"closer to being a practical alternative."**


* * *

### **Clarification of Common Concerns & New Results**

Our rebuttal addresses the key concerns raised by reviewers, supported by extensive new experiments.

-   **Concern 1: Scalability of Findings (Model Size).**

    -   **The Concern:** Reviewers (9dpA, cfK5, fm5B, rwo7) noted that our experiments were limited to GPT-2 small/medium scales (≤350M) and questioned if the findings hold for larger models.

    -   **Our New Experiment (774M Scaling; GPT-2 Large scale):** We took the initiative to train a **774M parameter AO-GPT** during the rebuttal. As detailed in our individual responses, the results demonstrate a **superior and consistent scaling trajectory**. The 774M model achieves significantly lower loss (2.98 vs 3.16 for 350M) and follows the predicted scaling law. This provides strong evidence that our findings regarding convergence and efficiency generalize to larger scales.


-   **Concern 2: The Role of Order Ensembling.**

    -   **The Misunderstanding:** Reviewers (cfK5) questioned the practicality of "Order Ensembling," concerned that it negates the speed advantage.

    -   **Our Clarification:** We clarified that ensembling was used **exclusively as a diagnostic tool** (Finding 6) to scientifically isolate the source of the perplexity gap (Order-Dependence vs. Invariance). **It is NOT a proposed inference strategy.**

    -   **Practical Implication:** For practical generation, AO-GPT operates with a **single permutation**, offering a Pareto trade-off: users gain a **~25× speedup** (Finding 8.1) in exchange for a moderate perplexity difference (~10-15%) compared to encoder-only models.


-   **Concern 3: The Role of 10% L2R Mixing.**

    -   **The Concern:** Reviewers asked if the 10% L2R mixing is a "band-aid" (cfK5) and how sensitive the model is to this ratio (fm5B).

    -   **Our New Experiment (Ablation Study):** We conducted a new ablation study comparing 10%, 50%, and 90% L2R mixing ratios. The results reveal a clear "sweet spot" at 10%. Increasing the L2R ratio further actually **degrades** any-order performance as the model over-specializes. This confirms that 10% L2R acts as a critical **inductive bias** to align optimization, rather than merely masking a flaw in the objective.


-   **Concern 4: Fairness of the Comparison.**

    -   **The Concern:** Reviewers (cfK5, rwo7) asked if comparing a "tuned" Decoder to an Encoder is fair, or if we should have adapted AR to an Encoder architecture.

    -   **Our Clarification:** We explained that the "tuning" (AdaLN/EMA) consists of standard diffusion practices necessary for the objective to work, not unfair tricks. Furthermore, adapting AR to an Encoder-only architecture (e.g., ENTP) is computationally infeasible for fair pre-training comparison due to the lack of causal masking (requiring $N$ forward passes per sequence with length $N$). Our setup remains the only computationally viable way to decouple formulation from architecture.

---

### Meta-Review · Area_Chair_yc3h · 2026-01-06

**Summary:**

This paper is a study on masked diffusion models. It aims to explain the performance gap between MDM and standard AR-GPT model. It points out that MDM differs from AR-GPT in both formulation and architecture, making the comparison complicated. In this paper, the authors try to decouple these two factors by introducing an Any-Order AR model that shares a similar backbone with AR model. The authors then design experiments to compare the performance of AO-AR and AR decoder-only models in Section 3 and performance of AO-AR encoder-only and decoder-only model in Section 4. Findings are reported based on these experiments.

**Reviewer Concerns:**

All the reviewers agree the overall idea is new and interesting. However, all of them raise concerns about the experiments. One major criticism is on the fairness of the comparison study. There could be unfair design choices and hyper parameters tuning. Another criticism is on the scale of the experiments. All the experiments are conducted in rather small scale and it is unclear whether the findings would hold in practice. On the presentation side, some reviewers find the narrative is lacking of focus; there are too many findings. Also, the ensemble on context order section is out of place and the method presented in this section is not convincing. The manuscript can benefit from a substantial modification based on the reviewers’ comments.

**Reviewer Scores:**

no

---

### Decision · Program_Chairs · 2026-01-26

Reject